



# Impact of livestock activity on near-surface ground temperatures in Mongolia

Robin B. Zweigel[1,2], Avirmed Dashtseren[3], Khurelbaatar Temuujin[3], Anarmaa Sharkhuu[4], Clare Webster[1], Hanna Lee[5], Sebastian Westermann[1,2]

[1]Department of Geosciences, University of Oslo, Oslo, 0371, Norway
[2]Centre for Biogeochemistry in the Anthropocene, University of Oslo, Oslo, 0371, Norway
[3]Institute of Geography and Geoecology. Mongolian academy of Sciences, Ulaanbaatar, 15170 , Mongolia
[4]Department of Biology, National University of Mongolia, Ulaanbaatar, 14201, Mongolia
[5]Department of Biology, Norwegian University of Science and Technology, Trondheim, 7419, Norway

*Correspondence to*: Robin B. Zweigel (robinbz@uio.no)

**Abstract.** Grazing by livestock can alter the surface conditions at grassland sites, impacting the transfer of energy between the atmosphere and ground and consequentially ground temperatures. In this study, we investigate surface cover in summer and winter and measure ground surface temperatures over 14 months at sites in Central Mongolia that feature different grazing intensities (intensely and ungrazed) and topographic aspects (north- and south-facing). Overall, intense grazing leads to a substantially reduced vegetation cover, altered snow conditions and lack of surface litter accumulation. Comparing intensely grazed and ungrazed plots shows large seasonal differences in ground surface temperatures, with grazed plots being up to +5.1°C warmer in summer and -5.4°C colder in winter at a south-facing site. We find smaller seasonal differences of +1.4°C and -2.5°C between grazed and ungrazed plots at a north-facing site which receives less solar radiation and where differences in vegetation cover between open and fenced plots are smaller. For both aspects, the seasonal differences largely offset each other, with both a small net cooling and warming depending on effects in spring and autumn. Our study suggests that livestock management might be used to modify the annual ground temperature dynamics, possibly even influencing local permafrost dynamics.

## 1 Introduction

Livestock activity affects ground surface temperatures (GSTs) at grassland sites through grazing and trampling, which changes the surface cover and consequentially the exchange of energy between the atmosphere and ground surface (Shao et al., 2017). Intense grazing reduces vegetation height and density (Wang et al., 2022; Yan et al., 2018), inhibits formation of litter layers (Hou et al., 2020), and changes the accumulation of local snow cover (Yan et al., 2019). Trampling by livestock compacts snow and ground surface layers, which changes their thermal properties (e.g. Gan et al., 2012; Zimov et al., 2012). At grazed grassland sites, these changes in surface cover lead to an amplification of the seasonal cycle of GSTs with distinctly higher GSTs in summer and lower GSTs in winter (e.g. Hou et al., 2020; Odriozola et al., 2014; Yan et al., 2018; Zhao et al., 2011). Furthermore, soil moisture levels are altered by intense grazing, yielding generally drier soils (Yan et al., 2018; Zhao et al., 2011) and reduced freeze-thaw processes (Wang et al., 2023). However, most studies linking grazing and

GSTs are limited to the growing season (e.g. Hou et al., 2020; Odriozola et al., 2014), or use controlled experiments where specified livestock loads are applied on fenced plots (Wang et al., 2023; Yan et al., 2018, 2019; Zhao et al., 2011). In reality,
grazing intensities are not spatially and temporally homogeneous, but determined by livestock and herders seeking optimal feeding conditions within their rangeland. The evaluation of the impacts of such gradients in grazing intensity on year-round GSTs has been limited.

In Mongolia, semi-nomadic pastoralism is widespread and more than 80% of the land area is used as rangeland for livestock
(Angerer et al., 2008; Fernández-Giménez et al., 2018). Mongolia's shift to market economy, growing population and high demand for cashmere wool have led to an increase and change in type of livestock over recent decades (Lkhagvadorj et al., 2013; Saizen et al., 2010). This intensified land use occurs together with intense climatic warming and has contributed to widespread land degradation in Mongolia (Wang et al., 2019, 2020). At the same time, Mongolian grasslands are often underlain by permafrost that is currently degrading (Ishikawa et al., 2018; Wang et al., 2022). Topography controls the local
distribution of vegetation cover and permafrost, which are both strongly linked to incoming solar radiation (Klinge et al., 2021; Zweigel et al., in press). So far, the impact of grazing and trampling on ground thermal dynamics in Mongolian grasslands and the relationship with topography has received little attention.

In this study, we investigate how differences in livestock activity in central Mongolian grasslands affect vegetation, snow
cover and associated GSTs. We measured GSTs at two sites with different topographic aspects and surveyed snow and vegetation cover in winter and summer. Based on these field measurements, we investigate the following questions:
1. How does livestock grazing and topography affect vegetation and snow cover in these Central Mongolian steppe environments?
2. How are seasonal and annual GSTs affected by differences in vegetation cover and litter amount induced by grazing and
trampling?

## 2 Study area

We collected data over a period of 14 months at two field sites; Terelj and Udleg, located ca. 50 km apart in the Khentii Mountains in Central Mongolia (ca. 48°N 107°E). The climate of the region is characterized as continental and generally dry, with large annual variations in temperature. Both sites are located in grasslands within the forest-steppe ecotone, where
topographic aspect and solar radiation produce distinct, local variations in vegetation cover (Klinge et al., 2021). In general, north-facing slopes feature boreal forest, while valley bottoms and south-facing slopes are dominated by grasslands (Ishikawa et al., 2005). The grasslands in this region experience extensive grazing from local livestock, which in both Terelj and Udleg consist predominantly of cattle, horses, yaks, goats and sheep.



The Terelj study site is located at 1650 m a.s.l. on a south-facing slope, in proximity to an automatic weather station described in Dashtseren et al. (2014) (Figure 1a). Dashtseren et al. (2014) report the local grassland on the south-facing slope to be permafrost-free with daily average air temperatures ranging from more than 20°C in summer to less than −30°C in winter, and the annual rainfall is less than 300 mm (2004-2007). At this intensely grazed site, sparse vegetation covered only about 60% of the ground surface in 2004, consisting of *Artemisia frigida*, *Potentilla acaulis*, *Agropyron cristatum* and *Carex*

*duriuscula* (Dashtseren et al., 2014). In 2003, the area around the weather station was fenced which created a sharp difference in livestock activity, as vegetation within the enclosure was fully protected from grazing and animal trampling (Figure 2). This allowed grass vegetation to grow tall within the fenced enclosure each summer, while the growth of vegetation in the surrounding open grasslands is limited by intense grazing.

Our measurement efforts in Terelj target the strong contrast in vegetation and livestock activity related to the fenced weather station enclosure. We established two field plots in the immediate vicinity of the fence, referred to as the "open" and "fenced" plots (Fig. 1c), where GSTs were monitored from May 2022 to August 2023. At the fenced plot, we limit interference from other activity associated with the weather station by focusing our field observations to an area in the north-western part of the enclosure, away from the gate and instrumentation. The open plot borders the fenced plot and is located

away from established animal and car tracks.

In Udleg, the studied grassland site is located on a gentle northeast-facing slope, situated at the base of a steeper forested north-facing slope (Figure 1b). The site is located at 1250-1290 m a.s.l., about 300 m lower in elevation than the Terelj site. From 2010-2012, meteorological and ground thermal data are available from a forest research tower in a boreal forest stand

(Miyazaki et al., 2014) ca. 500 m away from our grassland sites (Figure 1b). The meteorological conditions in Udleg are similar to those in Terelj, with monthly air temperatures ranging from -20 to -25°C in winter to +15 to +20°C in summer and annual rainfall between 230 and 317 mm. Miyazaki et al. (2014) found warm permafrost below the forest in Udleg, but the ground thermal state below the nearby grassland site is unknown. Vegetation surveys in Udleg in 2014 and 2015 show *Asteracaeae*, *Fabaceae* and *Caryophyllicaea* to be the most abundant grassland vegetation (Lkhavgadorj et al., 2016). The

research station and nearby grasslands are surrounded by a network of fences, which restrict livestock activity and thus create a gradient of grazing intensities at this site. For this reason, the upper parts of the slope are largely protected from livestock, while the lower part outside the fence system is intensely grazed. During the first part of our measurement period (Sect. 3.1) a breach in an outer fence allowed for low-intensity scattered grazing on the upper, southern part of the slope.

In Udleg, we establish three field plots aimed to capture the gradient in livestock activity (Figure 1d). This includes an "open" plot on the intensely grazed lower part of the slope and two fenced plots ca. 400 m away in the upper part of the slope. These latter plots target the difference in vegetation on each side of the fence between the two enclosures, where scattered grazing occurred in the outer part ("fenced A") due to a breach in the outer perimeter. This breach was repaired in



spring 2023, and no livestock were allowed within fenced plot A during summer 2023. On the other side of the fence, there
is a smaller enclosure that was protected from grazing in both summer 2022 and 2023 ("fenced B"). However, livestock was
herded into fenced plot B for a period in winter 2023, leading to substantial trampling and grazing of standing litter.

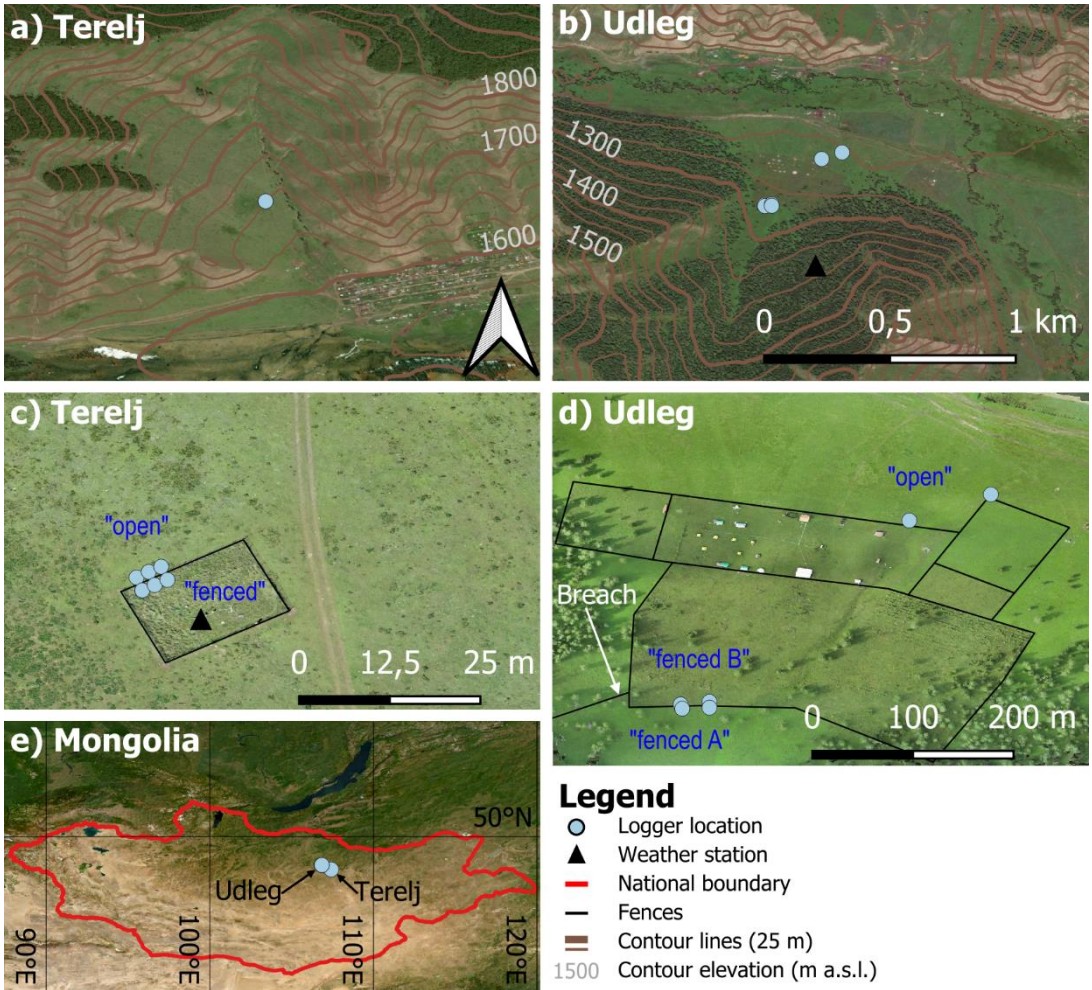

**Figure 1: Overview showing the location of the temperature loggers in Terelj (a & c) and Udleg (b & d), and the location of our**
**study sites within Mongolia (e). The location of our plots within each site is indicated by blue text in c) and d), while the white**
**arrow shows the location of the fence breach in 2022. All figures are oriented with North facing up, and a) and b) have the same**
**scale. Background: a), b) and e) Satellite imagery (ESRI, 2023), c) and d) drone-based orthomosaic acquired on 17. and 15. August**
**2023, respectively, and processed in Agisoft Metashape (Agisoft LLC, 2023).**





# 3 Methods

## 3.1 Ground surface temperature monitoring

In this study, we used iButton (©Maxim) temperature loggers that were installed at the different plots in summer 2022 (Table 1). These loggers are small (ca. 1.5 x 0.3 cm) and can be inserted at 2-3 cm below in the ground surface with minimal disturbance to vegetation and soil matrix. We use iButton loggers (DS1925 and DS1922L) that feature an accuracy of 0.5°C, a numerical resolution of 0.0625°C, and an operation range from -40°C to + 85°C (Maxim Integrated, 2024). For this study, we compute daily average GSTs from measurements at 4-hour intervals.

The logger placement aims to capture the variation in livestock activity and terrain among the plots. In Terelj, we deploy three logger-pairs (numbered 1-3 from west to east, Fig. 1c), each consisting of two loggers that are placed 2 m away from the fence on each site. This setup allows for redundancy in case of logger failure, while also capturing some of the spatial variability within the open fenced and fenced plots. In Udleg, we first placed two logger-pairs along the fence between fenced plot A and B, with each logger 3 m from the fence. Later in summer 2022, we supplemented loggers at two more locations in the open plot ca. 400 m away. The individual logger locations in Udleg are named "west" and "east", based on their relative location within each plot (Figure 1d). For all logger locations we derived local terrain parameters from the GLO-30 Digital elevation model (DEM) by ESA & Sinergise (2021). Furthermore, we quantify the differences in solar radiation among our plots by calculating the mean annual incoming solar radiation as in Zweigel et al. (in press), which considers the effects of both local slope and blocking by surrounding terrain (Table 2).

We calculated seasonal and annual average GSTs for all loggers with sufficient data. To ensure direct comparability among mean annual GSTs (MAGST), we calculated these over the 365-day period ending on 13 August 2023, which is contained in all logger measurements. For comparison of seasonal temperature signals we averaged daily GSTs for three-month periods starting from June 2022 to August 2023. As loggers were placed and read out during summer, we did not obtain a complete record for the summer season (June, July, and August) for all loggers. For the analysis, we still provide an estimate of the mean GST for summer for plots where daily GSTs are available for at least two thirds of the days.

**Table 1: Overview of ground surface temperature measurements used in this study. Coordinates are obtained from phone GPS. \* Indicates loggers that are placed substantially closer than the resolution of the DEM (30m) and are thus considered to have the same position and terrain parameters in the further analysis.**

| Logger ID [«site»_»plot»_»logger»] | Time period | Coordinate |
|---|---|---|
| Terelj_fenced_1 | 29. May 2022 – 13. August 2023 | 47.9897°N, 107.4221°E * |
| Terelj_fenced_2 | 29. May 2022 – 13. August 2023 | |
| Terelj_fenced_3 | 23. July 2022 – 13. August 2023 | |





| | | |
|---|---|---|
| Terelj_open_1 | 29. May 2022 – 13. August 2023 | |
| Terelj_open_2 | 29. May 2022 – 13. August 2023 | |
| Terelj_open_3 | 23. July 2022 – 13. August 2023 | |
| Udleg_fenced_A_east | 24. June 2022 – 15. August 2023 | 48.2598°N, 106.8488°E * |
| Udleg_fenced_B_east | 24. June 2022 – 15. August 2023 | |
| Udleg_fenced_A_west | 24. June 2022 – 15. August 2023 | 48.2597°N, 106.8484°E * |
| Udleg_fenced_B_west | 24. June 2022 – 15. August 2023 | |
| Udleg_open_east | 15. July 2022 – 14. August 2023 | 48.2625°N, 106.8529°E |
| Udleg_open_west | 15. July 2022 – 14. August 2023 | 48.2622°N, 106.8512°E |

**Table 2: Terrain parameters for the GST measurements locations as derived from the GLO-30 DEM. Aspect is given in degrees**
**counterclockwise from south following the right-hand notation by Dozier & Frew (1990), with compass directions indicated in brackets. SVF: Sky View Fraction. Sin: mean annual incoming solar radiation, calculated over the same period as MAGSTs (Sect. 4.3) using ERA5 reanalysis data (Hersbach et al., 2020) and the terrain adjustment methodology in Zweigel et al. (in press).**

| Field site | Plot(s) | Loggers | Altitude [m a.s.l] | Slope [°] | Aspect [°] | SVF [-] | Sin [W/m²] |
|---|---|---|---|---|---|---|---|
| Terelj | Fenced | 1, 2, 3 | 1651 | 7.5 | 57 (SE) | 0.91 | 177 |
| | Open | 1, 2, 3 | | | | | |
| Udleg | Fenced A, B | East | 1283 | 5.5 | 130 (NE) | 0.94 | 157 |
| | Fenced A, B | West | 1286 | 7.2 | 120 (E-NE) | 0.94 | 157 |
| | Open | East | 1251 | 3.6 | 124 (NE) | 0.94 | 162 |
| | Open | West | 1258 | 4.0 | 150 (N-NE) | 0.94 | 159 |

## 3.2 Additional field observations

We assessed the surface cover at the fenced and open plots in Terelj and Udleg in summer of 2022 and 2023, as well as winter 2023. During the growing season, we measured the typical vegetation height and took hemispherical photographs at the ground surface from which vegetation density was estimated (see below). We also noted the occurrence and thickness of intact litter layers at the ground surface, and the bare soil fraction, i.e. the surface fraction not covered by vegetation. For consistency, we use the term "vegetation" for live plants, whereas "litter layers" or "standing litter" is used respectively for

intact, dead plants that are standing upright or forming a largely horizontal layer near the ground surface. During the winter field campaign, we surveyed snow conditions with focus on spatial variability due to livestock activity. In Terelj, we also conducted a more comprehensive survey of vegetation and snow cover in winter and summer of 2023 to capture the small-scale spatial variability.





We used the hemispherical photographs to estimate leaf area index (*LAI*) of the grassland vegetation. The concept is to distinguish between pixels containing vegetation and sky across the 180° hemisphere up-looking perspective and calculate the leaf area per unit ground area that would produce such a fractional distribution of vegetation gaps. While such *LAI* calculations are subject to assumed vegetation properties and are sensitive to lighting conditions and exposure (Thimonier et al., 2010; Zhang et al., 2005), they provide a quantitative measure of vegetation density that can be compared among plots.

In this study we estimated *LAI* with the software Hemisfer (Schleppi et al., 2007) using the methodology by Lang (1987). An overview of the processing of hemispherical photographs is provided in Appendix A.

At each plot, we measured snow depths and densities in winter 2023. The number of measurements per plot depended on the apparent variability of the snow cover at the time of visit. We determined the column-average density of snow from the mass

of snow sampled by a cylinder of known dimensions that was vertically inserted into the snowpack. For sites featuring a clear layering, we also determined the density for the dominant layers. Furthermore, at each plot we observed the dominant snow grain sizes and litter content within the snow layers.

## 4 Results

### 4.1 Summer – vegetation

#### 4.1.1 Terelj

In the Terelj site we find strong contrasts in vegetation height and density between the open and fenced plots (Figure 2). In late May 2022, the open plot featured only low grass vegetation, with a bare soil fraction around 50%. At the same time, the ground at the fenced plot was completely covered by a thick litter layer, with no live vegetation visible (Figure 2a). By July 2022, a thick vegetation cover had established at the fenced plot, while the ground below was still covered by a litter layer.

At this time the grass at the open plot was still low with a high bare soil fraction (Table 3), except for scattered patches of *Artemisia frigida* (Figure 2c). A similar vegetation cover was found in August 2023 (Figure 2b, d &Figure 4a), where a survey of 30 points at the open plot showed a median vegetation height of 3 cm and 10% of the area covered by ca. 20 cm tall *Artemisia frigida* (Figure 3a). This survey also showed that large parts of the ground surface at the open plot were still exposed to the atmosphere (Figure 3b). A similar survey of 30 points at the fenced plot showed that while standing

vegetation was ca. 40-45 cm tall, the median height was 20 cm due to large parts of the grassland vegetation being blown over at this time (Figure 2d, Figure 3a & 4b-c). The survey at the fenced plot also showed that the ground was covered by a thick litter layer with a median thickness of 6 cm (Figure 2b & Figure 3c).



**Figure 2: The evolution of vegetation and ground cover at the Terelj site throughout the summer season. a) 29. May 2022: The open plot featured low grass vegetation while the fenced plot was covered by a thick litter layer. b) 17. August 2023: Example of the litter layer which at all times covered the ground surface in the fenced plot. c) 16. July 2022: A continuous vegetation cover of 30-50 cm had formed at the fenced plot, while the open plot is covered mostly by low vegetation. d) 17. August 2023: The open plot still featured sparse vegetation, while large fractions of the vegetation at the fenced plot had fallen/blown over.**




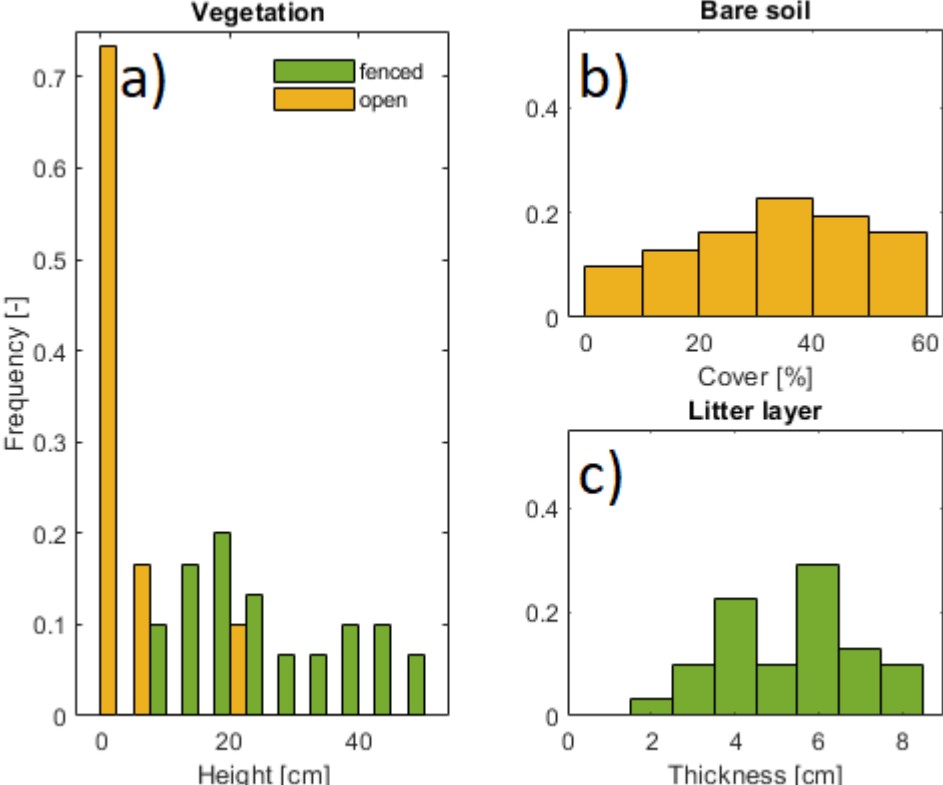

**Figure 3: Observations from the vegetation survey in the Terelj site on 17. August 2023 (n=30 in each plot). a) Distribution of vegetation height in the open (median = 3cm) and fenced plot (median = 20 cm). b) Distribution of bare soil fraction at the open plot (median = 30%). c) The thickness of intact litter layers covering the ground surface at the fenced plot (median = 6 cm). For readability, the bars in a) are plotted next to each other within each 5 cm interval.**

The *LAI* estimates differ considerably between the open and fenced plots in Terelj, showing a similar pattern as vegetation height (Figure 4a-c and Table 3). We found that throughout the summer, *LAI* at the open plot is at least one order of magnitude smaller than at the fenced plot.

**Table 3: Estimated LAI and vegetation height for the plots in Terelj from field visits in summer 2022 and 2023. LAIs are the mean of observations listed in Table A1. *These vegetation heights are the median of the survey presented in Figure 3a. n.m.: not measured.**

|  | Open: |  | Fenced: |  |
| --- | --- | --- | --- | --- |
| Date | LAI | Vegetation height | LAI | Vegetation height |
| 29. May 2022 | 0.01 m$^2$/m$^2$ | n. m. | 2.90 m$^2$/m$^2$ | n. m. |
| 16. July 2022 | 0.37 m$^2$/m$^2$ | n. m. | 3.57 m$^2$/m$^2$ | 30 cm |
| 23. July 2022 | 0.10 m$^2$/m$^2$ | 2 cm | 4.35 m$^2$/m$^2$ | 30 – 50 cm |
| 13./17. August 2023 | 0.10 m$^2$/m$^2$ | 3 cm* | 1.81 m$^2$/m$^2$ | 20 cm* |




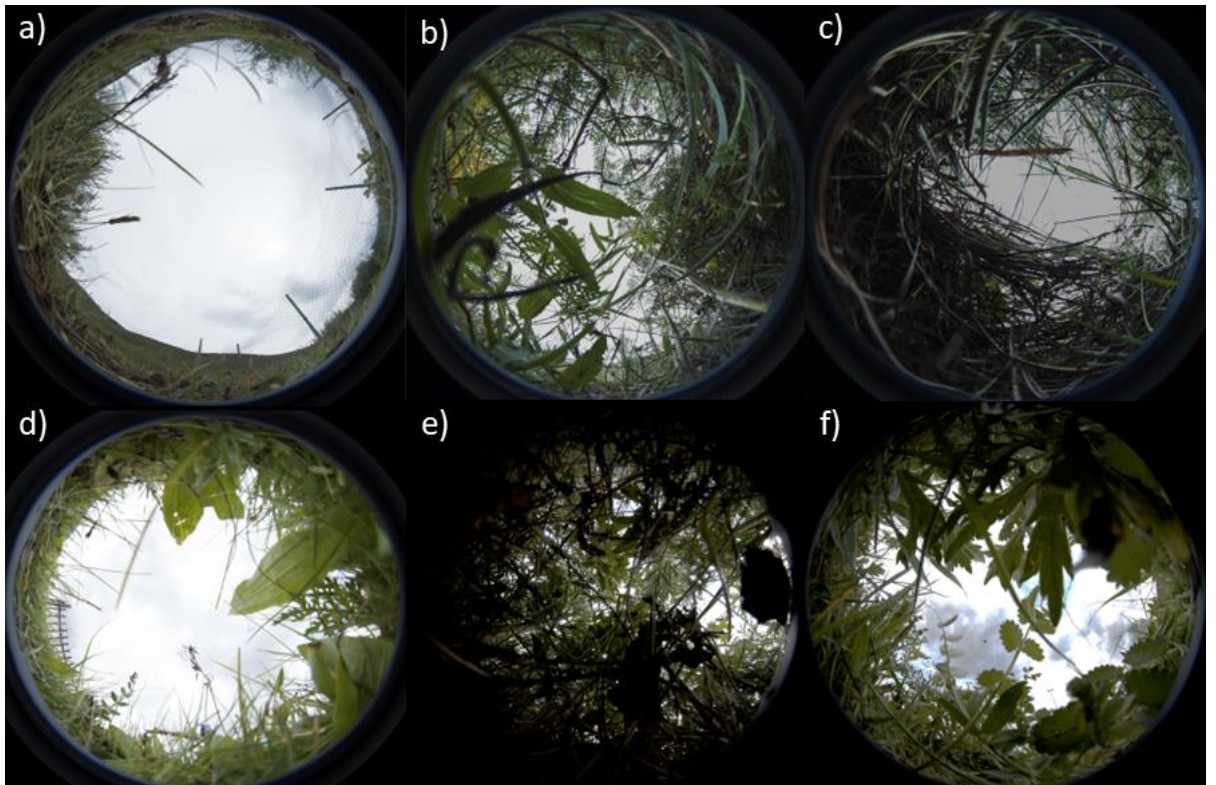

**Figure 4: Example of hemispheric photographs used to derive LAI. Terelj: a) 13. August 2023, open plot (LAI=0.19), b) 13. August 2023, fenced plot (LAI=2.10), c) 13. August 2023, fenced plot with wind-blown vegetation (LAI=1.77). Udleg: d) 16. August 2023, open plot (LAI=0.77), e) 18. July 2022, fenced plot B (LAI=4.65), f) 18. July 2022, fenced plot A (LAI=2.71).**

### 4.1.2 Udleg

The three plots in Udleg feature a gradient in vegetation cover (Figure 5). Vegetation heights were lower in fenced plot A compared to fenced plot B in June and July 2022, which is likely due to a fence breach that allowed for scattered grazing at
fenced plot A. This difference in vegetation height was maintained even after the fence breach was repaired in spring 2023 (Table 4 and Figure 5c). Contrary to the fenced plot in Terelj, we only observed shallow litter layers in Udleg, measuring 0.5 and 1 cm at fenced plots A and B, respectively, in August 2023. The vegetation height at the open plot in Udleg was 5 cm in both July 2022 and August 2023 (Table 4), but was denser than at the open plot in Terelj and completely covered the ground surface (Figure 5b &Figure 4d).


The LAI-based estimates of vegetation density at Udleg provide additional insight to the vegetation height measurements (Table 4). When fenced plot A was subject to scattered grazing in 2022, it had a considerably lower *LAI* than the ungrazed fenced plot B. However, in 2023, when both fenced sites were protected from grazing, fenced plot A reached a somewhat



higher *LAI* than fenced plot B. We note that this was contrary to the observed vegetation heights, which still were markedly

higher at fenced site B.

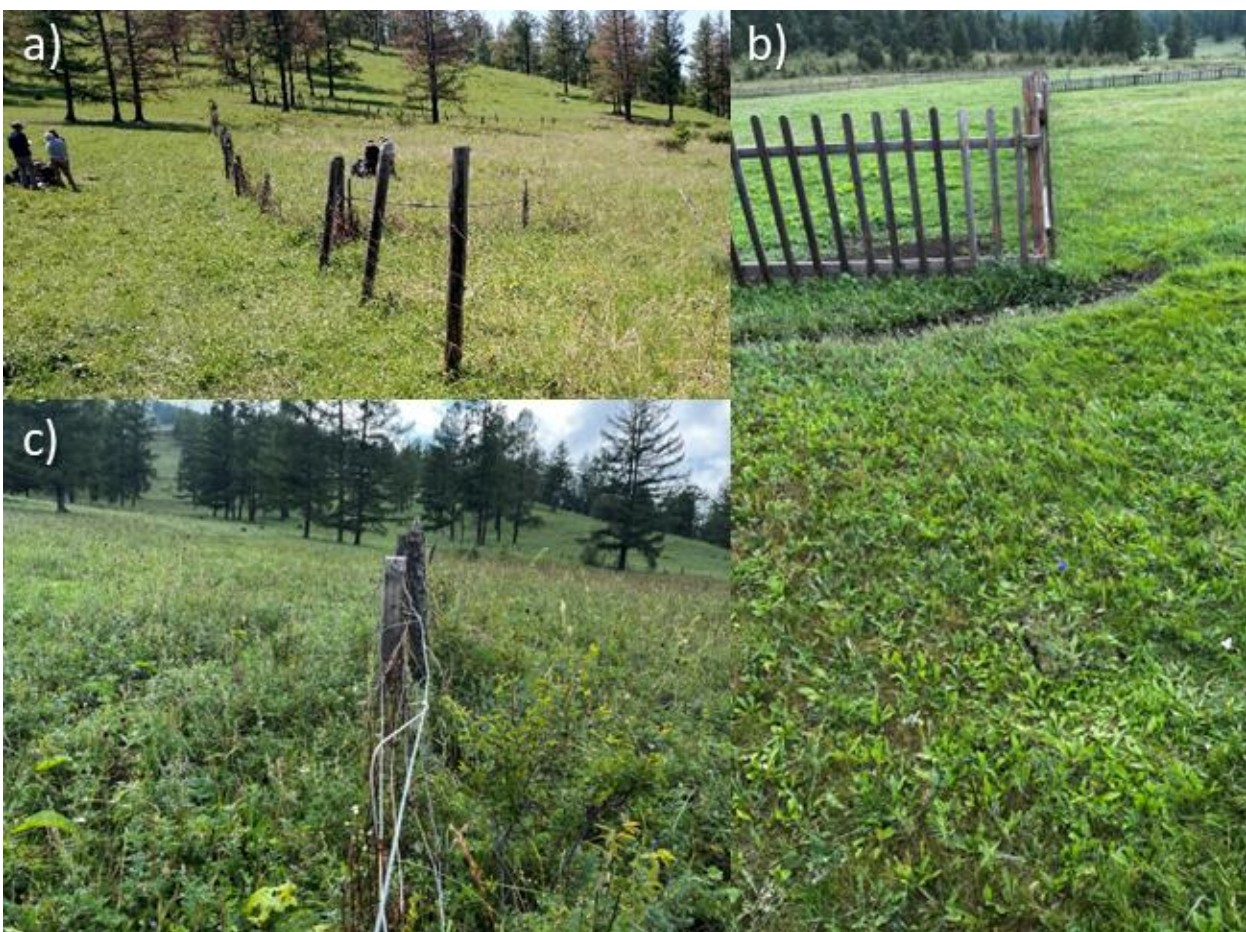

**Figure 5: Vegetation cover at the plots in the Udleg study area. a) June 24. 2022: A fence breach led to sparse grazing and lower vegetation heights left of the depicted fence (fenced plot A). There is higher vegetation right of the fence where no grazing animals have access (fenced plot B). b) 18. July 2022: the open plot is characterized by intense grazing and low vegetation. c) 14. August 2023: After repairing the fence in spring 2023 the vegetation density is more similar in fenced plots A and B.**






**Table 4: Estimated LAI and vegetation height for the plots in Udleg from field visits in summer 2022 and 2023. LAIs are the mean of observations listed in Table A1.**

| | Open: | | Fenced A: | | Fenced B: | |
| Date | LAI | Vegetation height | LAI | Vegetation height | LAI | Vegetation height |
| --- | --- | --- | --- | --- | --- | --- |
| 24. June 2022 | n.m. | n.m. | 0.95 $m^2/m^2$ | n.m. | 2.43 $m^2/m^2$ | n.m. |
| 18./19. July 2022 | 0.14 $m^2/m^2$ | 5 cm | 3,20 $m^2/m^2$ | $20-30$ cm | 4.55 $m^2/m^2$ | 50 cm |
| 14./15. Aug. 2023 | 1.67 $m^2/m^2$ | 5 cm | 3.27 $m^2/m^2$ | $20-25$ cm | 2.67 $m^2/m^2$ | $30-40$ cm |

## 4.2 Winter

### 4.2.1 Terelj

The snow cover in the fenced plot in Terelj was deeper than in the open plot and it was substantially affected by the presence of litter in late winter 2023 (Figure 6). Data from two snow pits at the fenced plot showed the snowpack to be 21-32 cm deep, with a clear layering due to a basal litter layer. The top layer of snow was 7-14 cm thick with a density of 215 $kg/m^3$ which was located on top of a 14-18 cm basal layer consisting of a mix of intact litter and snow. In this basal layer, the litter component provided structural support for the top snow layer, and the snow matrix was discontinuous and dominated by chains and pockets of depth hoar, with a snow density of 135 $kg/m^3$. In many places, standing litter also protruded through the snow surface (Figure 6).

In the open plot, the snow cover was rather shallow and did not show any clear horizontal layering. In the immediate vicinity of the GST logger locations, we measured 10-15 cm deep snow that did not show signs of disturbance due to trampling since the last snowfall. However, trampling due to livestock grazing was evident in the wider, open area, leading to the compaction or locally even the complete removal of snow cover. We investigated the effect of trampling on the variability of the snow depths by measuring 100 sites in two areas, one of which had experienced substantial trampling, while the second showed no signs of trampling since last snowfall. Figure 7Figure 7 shows that while the average snow depth in these areas was comparable, there is a stronger variability in snow depths in the trampled area. We also sampled column-average snow density at 25 locations in the visually undisturbed area, yielding an average snow density of 166 $kg/m^3$.



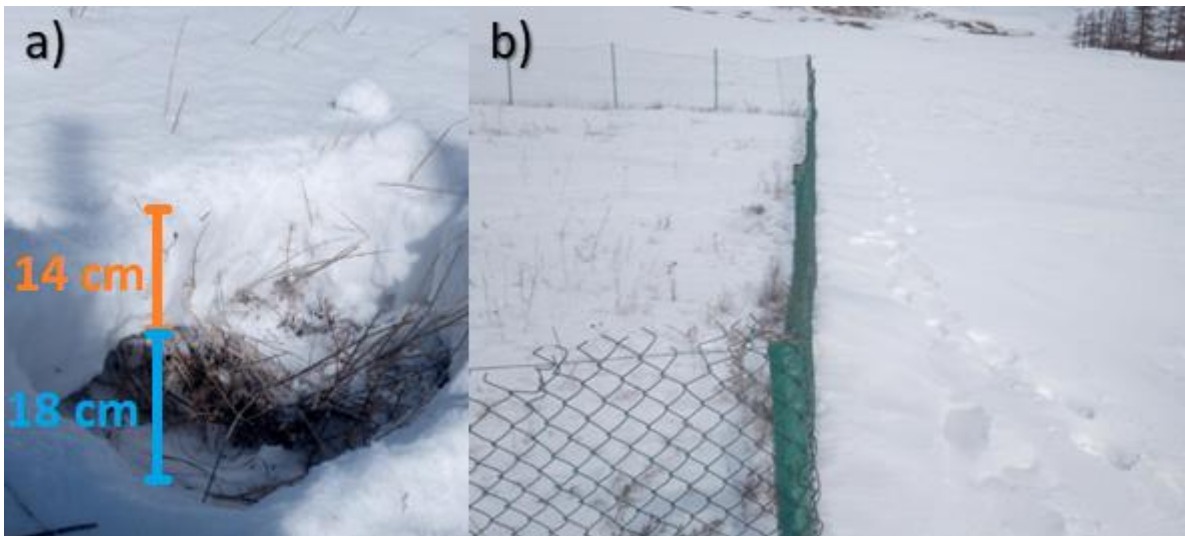

**Figure 6: Snow cover at the open and fenced plot in Terelj (27. February 2023). a) At the fenced plot, a denser layer of snow sits on top of a thick litter layer and depth hoar. b) Standing litter still protrudes through the snowpack at the fenced plot. Snow cover in the open plot is more continuous and characterized by local trampling.**

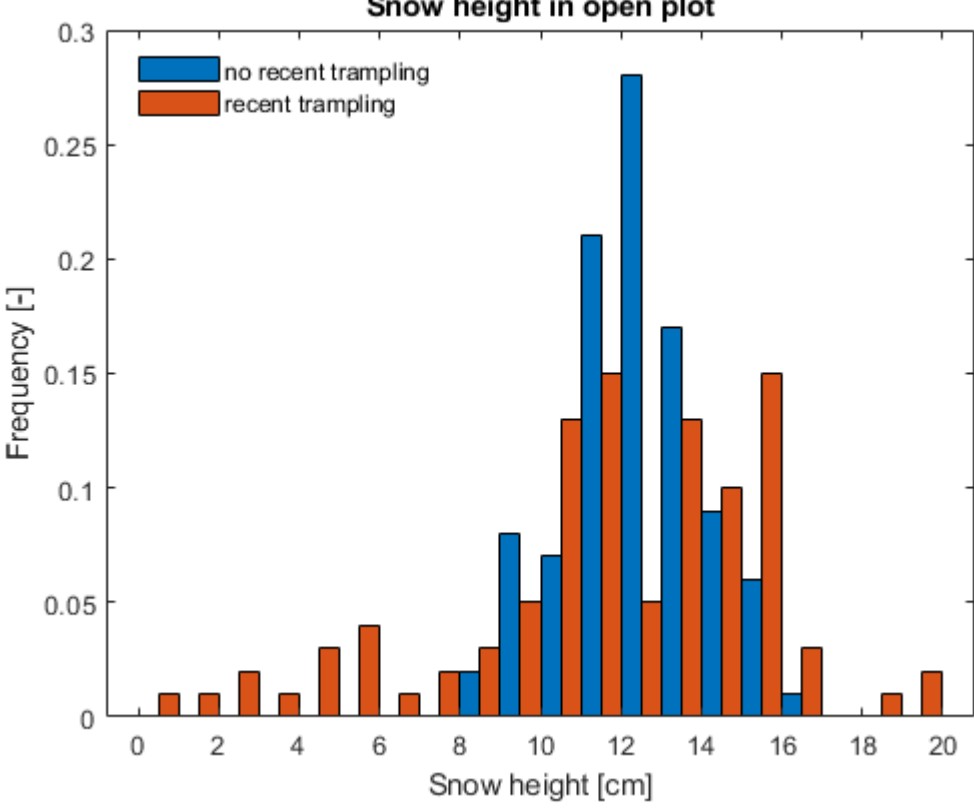

**Figure 7: Histogram of 100 snow depths sampled in areas with and without recent trampling in the open plot in Terelj (27. February 2023). For readability, the bars in a) are plotted next to each other within each 1 cm interval.**



### 4.2.2 Udleg

At Udleg, the snow cover is characterized by high spatial variability due to livestock trampling. Livestock had access to all plots in the period prior to our field visit in late February 2023 (Figure 8), with the most intense trampling visible at fenced plot B. Fenced plot B featured a large variability in both snow depth and density, but undisturbed locations in both fenced plot A and B showed snow depths of 20 cm or more (Figure 9Figure 9). While there was also scattered trampling at the open plot, no disturbances were observed in the vicinity of the logger locations at the time of our field visit. At the open plot, snow

densities were consistently around 180 kg/m³ while snow depths were consistently lower than undisturbed locations in the fenced plots (Figure 9Figure 9). The variability in snow depths in the open is likely linked to local wind redistribution of snow, consistent with observed wind transport features at the snow surface (Figure 8b). We also observe a basal layer with high content of standing litter in the snowpack at both fenced plots (Figure 10).

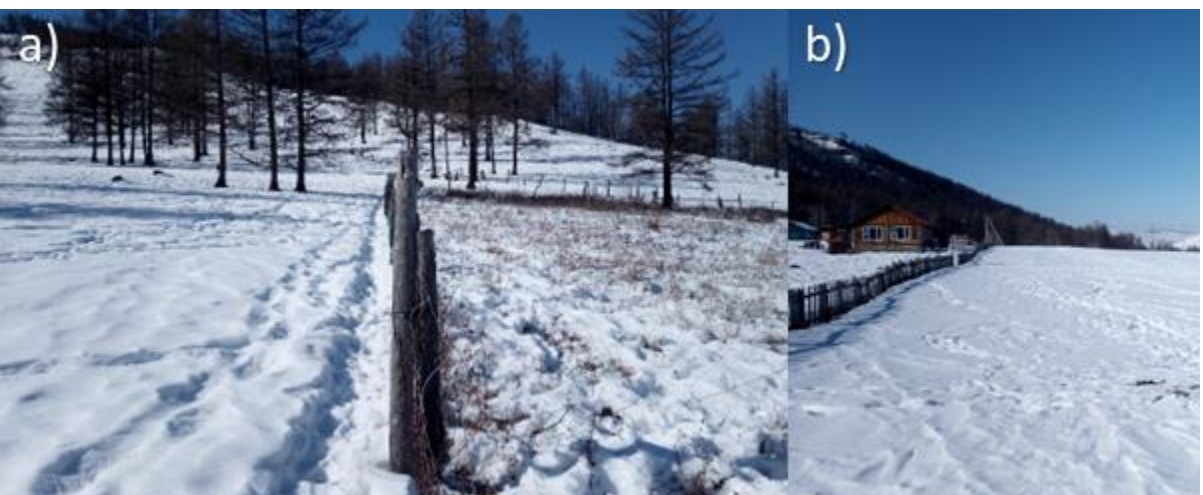

**Figure 8: Snow conditions in the Udleg study area on 25. February 2023. a) Fenced plot A (left of fence) has received less trampling than fenced plot B (right of fence). b) Open site with wind transport features and scattered livestock tracks.**




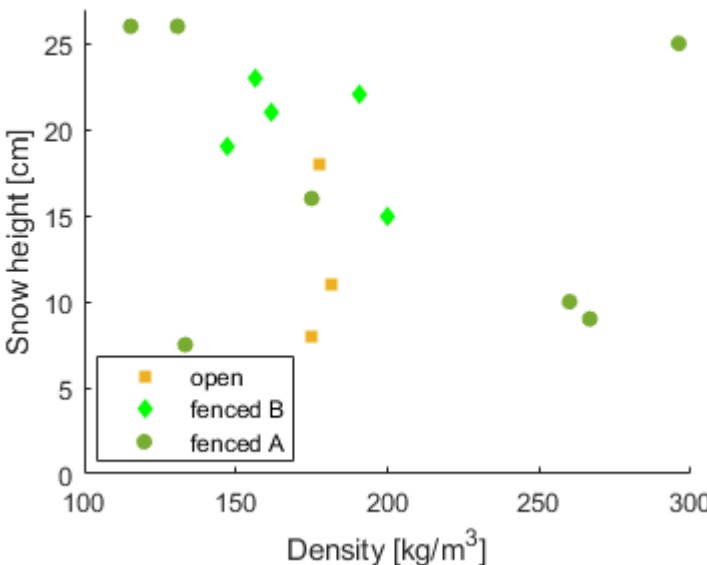

**Figure 9: Snow measurements from Udleg on 25. February 2023. Observations are taken in the immediate vicinity of the GST logger locations, showing the spatial variability of snow depth and densities in each plot.**

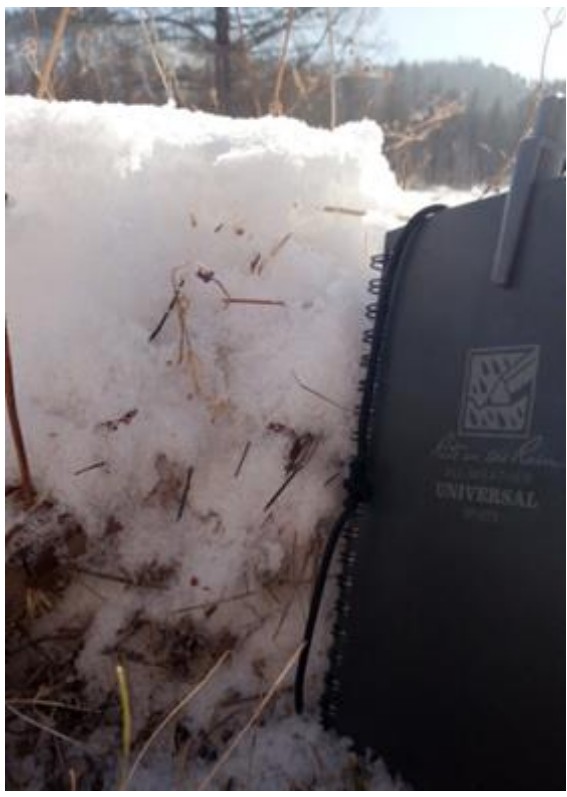

**Figure 10: Snow profile of an undisturbed location at fenced plot B, showing a large content of standing litter in the lower half of the snowpack. Notebook (18 cm) for scale.**



## 4.3 Ground surface temperatures

### 4.3.1 Terelj

The open and fenced plots in Terelj display different seasonal patterns in ground surface temperatures. Differences are largest between April and September when GSTs in the intensely grazed open are substantially warmer than at the ungrazed fenced plot, and from December to February when GSTs in the open are colder (Figure 11 Figure 12). Notably, the open and fenced plot feature rather similar GSTs for two months from the onset of freezing in September until the middle of November, after which GSTs are strongly different between the plots for the remainder of winter. This sudden deviation of

GSTs could be related to the start of significant livestock activity at the logger locations in the open plot which disturbed the snow cover. While the absolute differences in summer and winter GST are of a similar magnitude, the open plot features warmer GSTs also in spring and autumn (Table 5). Over the entire year, the open plot features a 0.7°C warmer MAGST than the fenced plot.

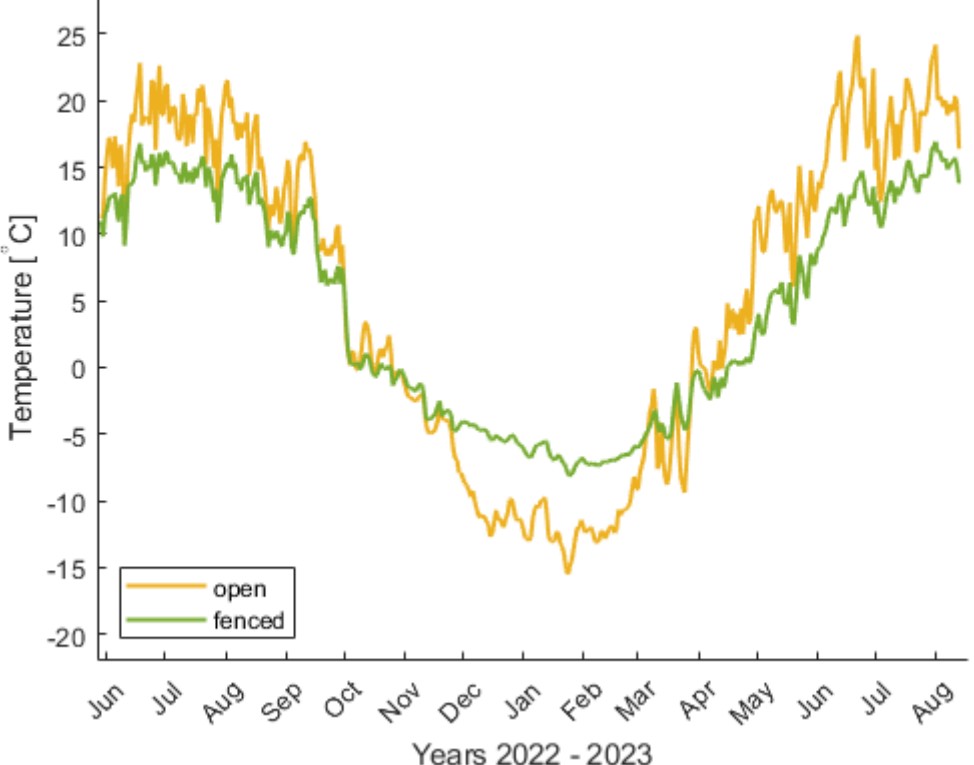

**Figure 11: Average daily ground surface temperature for the plots in Terelj. The GST evolution of the individual loggers is presented in Fig. B1.**





**Table 5: Annual and seasonal three-month average GSTs for the plots in Terelj. JJA = June, July & August, SON = September, October & November, DJF = December, January & February, MAM = March, April & May. MAGST is calculated over the 365 ending on 13. August 2023. Corresponding annual and seasonal GSTs of the individual loggers are presented in Table B1.**

| Plot | MAGST | JJA (2022) | SON | DJF | MAM | JJA (2023) |
|---|---|---|---|---|---|---|
| **open** | **3.1** | **17.2** | **2.9** | **-11.6** | **2.8** | **18.8** |
| **fenced** | **2.4** | **13.1** | **2.2** | **-6.2** | **0.4** | **13.3** |
| Difference (open – fenced) | +0.7 | +4.1 | +0.7 | -5.4 | +2.4 | +5.1 |

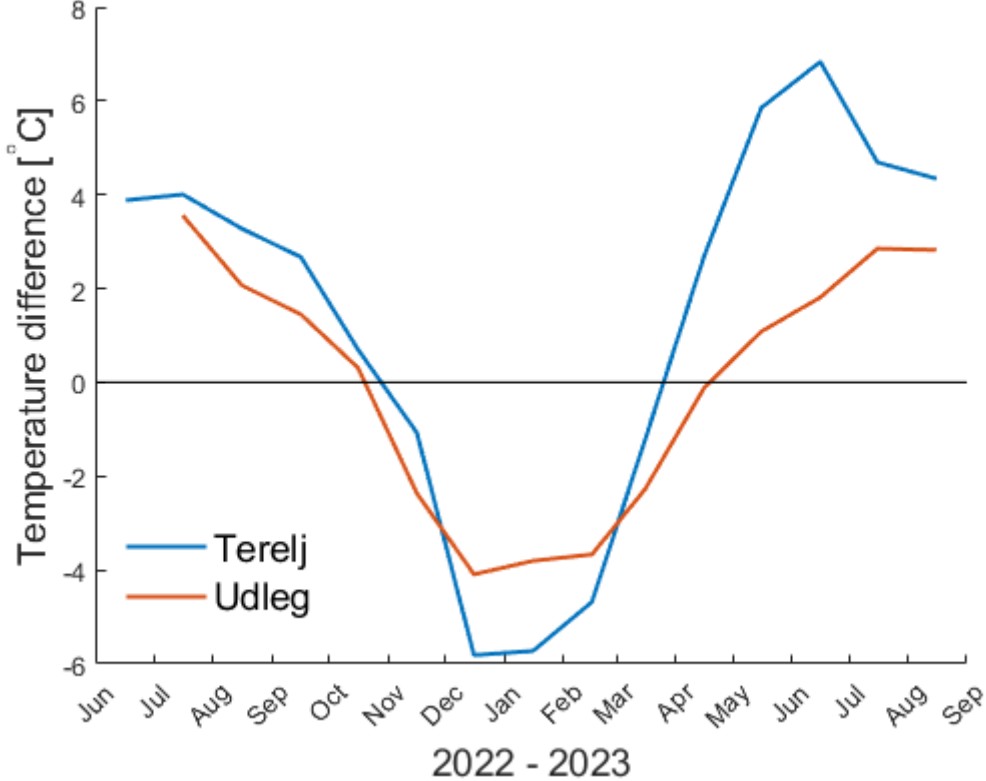

**Figure 12: Differences in monthly GST (mean of loggers in open plot – mean of loggers in fenced plots) in Terelj (south-facing) and Udleg (north-facing).**

### 4.3.2 Udleg

At Udleg, the differences in ground temperatures between open and fenced plots are less pronounced compared to Terelj (Figure 12). However, GSTs are still significantly warmer in the open plot in summer, but they are colder than in the fenced plots not only in winter, but also in autumn and spring (Figure 13 & Table 6). The largest difference in GST between the





open and fenced plots is observed in winter, while the differences in the transitional periods in spring and autumn are small. Overall, MAGSTs are lower in the open plot compared to the fenced plots, but this difference is smaller than the spatial

variation within each plot (Table B2). Similar to Terelj, we note a strongly deviating GST evolution for one logger in the open plot starting in December, which potentially is linked to the onset of snow disturbance by livestock (Fig. B2). Furthermore, we note that the annual and most seasonal GSTs are colder in Udleg compared to the corresponding plots in Terelj, despite Udleg being located at ca. 300 m lower elevation, which is likely caused by the difference in aspect.


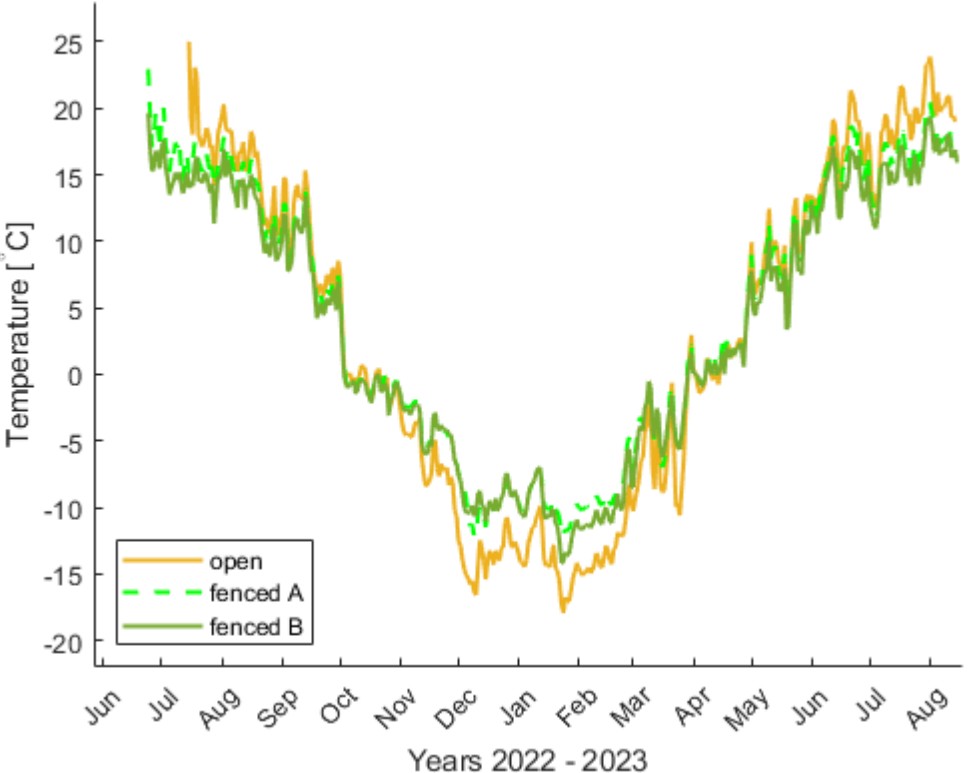

**Figure 13: Daily ground surface temperature from the loggers placed in Udleg. Note that GST are measured at fenced plot A and B from 24. June 2022, while the open plot was added on the 15. July 2022. The GST evolution of the individual loggers is presented in Fig. B2.**






**Table 6: As Table 5, but for the plots in Udleg. Note that the loggers in the open have insufficient measurements for JJA in 2022. Corresponding annual and seasonal GSTs of the individual loggers are presented in Table B2.**

| Logger | MAGST | JJA (2022) | SON | DJF | MAM | JJA (2023) |
|---|---|---|---|---|---|---|
| **open** | **1.7** | **-** | **1.2** | **-13.7** | **1.6** | **18.1** |
| fenced A | 2.5 | 15.4 | 1.6 | -9.5 | 2.3 | 16.2 |
| fenced B | 1.9 | 14.0 | 1.2 | -10.1 | 1.8 | 15.1 |
| **fenced mean** | **2.2** | **14.7** | **1.4** | **-9.8** | **2.0** | **15.7** |
| Difference (open –fenced mean) | -0.4 | - | -0.2 | -2.5 | -0.4 | +1.4 |

We also observe differences in annual and seasonal GSTs between the two fenced plots in Udleg (Table 6). Fenced plot A, which has lower vegetation height and density (Table 4), has higher GSTs than fenced plot B (Figure 13). This difference is consistent throughout the year, and plot A has average MAGSTs 0.6°C warmer than plot B (Table 6). The largest temperature difference between the two fenced plots is observed in the summer seasons, also in 2023 when both fenced plots were protected from grazing (Table 6 and Figure 13). However, fenced plots A and B feature a similar seasonal evolution of

GSTs, distinctly different from the open plot.

**5 Discussion**

In this study, we use measured GSTs and observations of grassland vegetation and snow cover from the Khentii Mountains in Central Mongolia to quantify and compare the different temperature regimes across contrasts of grazing intensity and topography. We find that the exclusion of grazing livestock allows for higher and denser vegetation cover (Table 3 & 4) and

a dampened seasonal cycle in GST (Figure 11 &Figure 13). We find the largest differences in GST at the south-facing Terelj site, where the grazed open plot is significantly warmer than the ungrazed fenced plot from April to September, while it is significantly colder in winter (Figure 12). At the north-facing Udleg site, we find differences in grazing intensity to produce less pronounced differences in vegetation cover and observe smaller seasonal differences in GST between open and fenced plots (Figure 13). Intense grazing has only a small, but contrasting effect on MAGSTs, with the open plot being 0.7°C

warmer than the fenced plot in Terelj, while it is 0.4°C colder in Udleg (Table 5 and Table 6). The difference in MAGST is small compared to differences in monthly GSTs, which range from -5.8°C colder to 6.8°C warmer at the south-facing Terelj site, and from -4.1°C colder to 3.6°C warmer at the north-facing Udleg site (Figure 12).

Our study complements previous research on the relationship between grazing intensity, vegetation cover and ground

temperatures. For example, a recent study found that removal of vegetation in cold climates can affect summertime GSTs through reduced shading (Jaroszynska et al., 2023). Vegetation removal is a main grazing effect, which we find to cause



higher summertime GST at grassland sites in semi-arid Central Mongolia. Furthermore, several studies have shown an intensification of the annual GST cycle with grazing intensity leading to several degrees higher GSTs in summer that are partly offset by colder wintertime GSTs (Yan et al., 2018; Zhao et al., 2011). However, most previous research linking 330 livestock activity and GST dynamics have used controlled experiments, and the effect of grazing gradients within a landscape dominated by pastoralism has received little attention. Our observations of GSTs and surface conditions under direct grazing effects thus represent a valuable dataset on how these effects manifest at sites with traditional livestock management.

### 335 5.1 Grazing impact on ground surface temperatures

Livestock grazing and trampling changes the surface properties at grassland sites, including reduced vegetation cover and compaction of ground, snow, and litter (e.g. Gan et al. (2012); Yan et al. (2018); Zimov et al. (2012)). These changes modulate the exchange of energy between the atmosphere and ground which drive the observed differences in GST between open and fences areas. The altered surfaces interact with other factors and processes, such as wind redistribution of snow and 340 terrain-driven differences in radiation. Here, we provide an overview of the processes relevant at our sites and relate them to observations of surface properties and GSTs.

Vegetation cover intercepts solar radiation, which leads to a lower net shortwave radiation at the ground surface. Such shading of the ground below lowers GSTs during the growing season (e.g. Yan et al. (2018)). Zhang et al. (2022) showed 345 that this shading effect increases with vegetation density and is greater for live than dead vegetation. Furthermore, Shao et al. (2017) found that although grazing reduced the net radiation above grassland sites due to a higher albedo, it also lead to a higher ground heat flux. These findings agree well with the dampened seasonal cycle and lower summertime GSTs observed at the fenced plots at the sites in Terelj and Udleg (Figure 11 & Figure 13).

We also find differences in GSTs between open and fenced plots outside the growing season, especially during late spring in Terelj (Figure 12). Here, the fenced plot warms considerably slower than the open plot, which could be linked to the insulation of the litter layers that were observed in summer. Such litter layers affect the ground surface thermal regime by intercepting radiation, limiting turbulent exchange, and lowering the ground heat flux (Yan et al., 2018). This reduced energy transfer can for example slow ground warming in spring, leading to a delayed plant phenology, increased soil moisture and 355 lower GSTs (Hou et al., 2020). While the exact role is unclear, litter layers could thus be an important mediator of GSTs at the Terelj site, especially in spring when all plots are snow free and grazing induced differences in vegetation have not yet become manifest.



Livestock activity affects the snow cover directly through trampling (see below), but also indirectly by inducing local
variations in snow ablation. Sublimation of snow can remove a substantial part of the available snow cover at continental
sites (Zhang et al., 2004, 2008), and the rate of sublimation depends heavily on the snow surface temperature and
consequently the surface energy balance. During our field visit in winter we observe snow-free conditions on steeper south-
facing slopes that receive high insolation in winter and early spring, and Sentinel-2 satellite imagery confirms that snow
cover on these slopes is of intermittent nature (ESA, 2024). Similarly, areas with trampling by livestock will feature a snow
microtopography where small south-facing surfaces are likely subject to increased sublimation, thus potentially reducing the
snow cover compared to undisturbed areas.

Wind redistribution of snow can be the main driver of snow depth variability within an area, which, especially at cold sites,
can drive large differences in wintertime GSTs (Zweigel et al., 2021). In the open plot at Udleg, wind is channelled through
the main valley, and we observe snow structures caused by wind drift near the measurement sites (Figure 8), as well as
deeper snow drifts in the surrounding. We also observe a snow cover with relatively homogeneous density, but variable
snow depths (Figure 9Figure 9), which could be explained by the effects of wind redistribution. Wind induced differences in
snow depth could thus be an additional factor causing the difference in wintertime GSTs, possibly in interplay with
vegetation removal by grazing. Yan et al. (2019) found that intense grazing can substantially reduce the snow accumulation
at grassland sites, which they link to the lowering of snow holding capacity due to lower vegetation. Such differences in
wind redistribution might also explain some of the differences in snow depth between the open and fenced plots at both sites,
where we observed the snowpack in the open plots to be thinner than in the fenced plots, which also feature substantial
standing litter within the snowpack (Figure 6 & Figure 10). Wind redistribution of snow further interacts with snow
trampling by grazing livestock by infilling depressions caused by grazing animals, thus again smoothing the surface. While
both these changes to the snow cover are event-based, trampling is a highly local and random effect while wind distribution
is spatially more consistent. Spatial differences in snow depth induced by livestock activity can be smoothed out by
subsequent wind drift events, which affects snow cover and densities. Overall, the effect of wind on snow cover and its
interaction with direct and indirect grazing effects must be considered at grassland sites when studying local variations in
wintertime GSTs.


Vegetation densities at grazed plots vary between our sites, which can be related to differences between south- and north-
facing slopes with respect to grazing intensity and soil moisture availability. Throughout our observations, the open plot in
Udleg features a continuous grass cover and higher LAI than the open plot in Terelj, which also has a high fraction of bare
soil (Figure 3b, 4a & d and Table 3 & 4). These variations in vegetation density could be linked to differences in livestock
activity, which can be induced by seasonal grazing patterns where e.g. south-facing slopes are preferentially grazed in
summer (see below). Alternatively, terrain induced differences in solar radiation (Table 2) can drive stronger evaporation on
south-facing slopes which can limited the soil moisture available for plant growth. For example, previous studies have



shown that evapotranspiration consumes all available precipitation and soil moisture on south-facing slopes in the Terelj area, while a surplus of soil water is available below forested north-facing slopes nearby (Iijima et al., 2012). Soil moisture

levels are also affected by recharge from snow melt, which in this region is affected by sublimation that consumes 20 – 29% of the annual snowfall at flat sites (Zhang et al., 2008). Zweigel et al. (in press) showed that gentle south-facing slopes in Terelj receive more solar radiation than north-facing slopes especially during winter, autumn and spring, which can drive a faster ablation of snow cover spring. With less snow water equivalent available for infiltration and an earlier snow melt out, a soil moisture deficit on gentle south-facing slopes could thus build up already prior to the main growing season.

Additionally, plant growth in north-facing slopes, such as Udleg, can be supported by meltwater inflow from local permafrost-underlain forest (Klinge et al., 2021). As gentle north- and south-facing slopes receive rather similar solar radiation in summer (Zweigel et al., in press), differences in summertime GSTs among the grazed plots in Terelj and Udleg are mostly related indirectly to terrain exposure through its impact on local vegetation density.

Whether grazing leads to higher or lower MAGSTs depends on the relative magnitude and timing of summer and winter effects. The Terelj and Udleg sites mainly differ in the incoming solar radiation (10-13% higher in Terelj; Table 2) and the amount of litter layers in the fenced plots. The higher MAGSTs in Terelj for both open and fenced plots, despite this site being located more than 300 m higher than Udleg, suggests that the solar radiation is a key driver of ground thermal regime at this south-facing site. As solar radiation is most intense in summer, the effects of shading vegetation during the growing

season dominate the annual GST signal at this site. The difference in GSTs in Terelj is further increased by the slower warming of the fenced site due to thick litter layers, with the largest effect in late spring (Figure 12). Overall, the Terelj data indicate warmer MAGSTs in response to intense grazing. In contrast, the Udleg site displays more similar GSTs across all plots in the warm season, and the net effect of grazing on MAGSTs depends largely on wintertime effects (Table 6). Here, local disturbances of the snow cover due to wind drift and livestock grazing seem to determine whether MAGSTs in the

open are higher or lower than those at fenced plots (Table B2).

Previous studies have suggested that snow trampling by livestock has a strong and decisive impact on the ground thermal regime, leading to generally colder ground temperatures (e.g. Zimov et al. (2012)). For Siberia, Zimov et al. (2012) report that herbivores are limited by the available feed during long winters, while grasslands can establish a substantial vegetation

cover during moderate grazing in summer. In such a system, vegetation can shade the ground from solar radiation during the growing season, while the snow cover in winter is subject to extensive trampling, giving overall cooling of ground temperatures. Our study region is located about 20° further south compared to the site studied by Zimov et al. (2012), and thus experiences higher amounts of solar radiation, as well as shorter winter and a longer and warmer growing seasons. Here, we observe intense grazing to limit the establishment of substantial vegetation cover in grazed areas in summer (Table 3 and




Table 4), leaving the ground directly exposed to the atmosphere especially on south-facing slopes (Sect. 4.1.1 Terelj). While
we did not observed direct trampling at our logger sites during the visit in winter, we observed that trampling leads to
increased variability of snow depths and densities (Figure 7 & 9), and we note sudden shifts in GSTs during winter for
loggers in the open plots (Terelj: mid-November (Figure 11), Udleg: start of December (Fig. B2)) which are highly likely
caused by livestock disturbance of the local snow cover. These grazed and trampled loggers experience the coldest
wintertime GSTs in Terelj and Udleg, respectively, but we do not observe a general cooling on annual basis (Table 5 and
B2). Notably, the disturbed loggers in Terelj feature higher MAGSTs than the undisturbed loggers in the fenced plot,
indicating that wintertime disturbances do not dominate the annual GST cycle at this site. Overall, the effects grazing has on
GSTs during the growing season seems to dominate the livestock impact on MAGST in our study region.

## 5.2 Limitations

Our study presents 14 months of GST measurements which is a limiting factor for the analysis of the interplay of grazing,
vegetation and GSTs. However, multi-year studies in grassland ecosystems show similar seasonal differences in GST as
found in this study, also for years with different precipitation conditions (Yan et al., 2018; Zhao et al., 2011). Thus, our
observations likely capture the key properties and dynamics of grazed and ungrazed grassland for these latitudes.

The complimentary field observations of surface conditions are limited to summer and late winter, which limits our
investigation of the causal relationships between livestock activity, ground surface conditions and GST at other times. The
presence of reflecting and insulating snow cover or litter layers will significantly impact the rates of ground warming and
cooling. For example, we note strong deviations in wintertime GST in some plots that are likely linked to disturbance of
surface cover (Figure 11 & B2) but lack observations for direct attribution of these events to livestock trampling. However,
Sentinel-2 satellite imagery shows a well-established snow cover at both field sites at these times (since 31. October at both
sites; ESA (2024)), and the deviating GSTs are thus likely linked to local disturbances of snow cover. Similarly, our
interpretation of litter layers as the main cause of delayed ground warming in spring in Terelj remains uncertain without
information of surface conditions at that time. To investigate the role of grazing outside the growing season, future studies
should include observations of surface conditions in spring, autumn and early winter.


Our study sites represent differences in livestock activity where the grazing intensity is not controlled, but rather a product of
local livestock and herders search for optimal feeding conditions. For example, the fenced plot B in Udleg was subject to
substantial trampling and winter grazing, and we measured a large variability of snow depth and densities at this plot in late
winter (Figure 8 & 9Figure 9). However, we do not observe different GST between fenced plot B and the visually



undisturbed fenced plot A, neither prior nor after our field visit (Figure 13). It is unclear whether this is because GSTs are rather insensitive to variations in snow cover towards the end of the cold season, or if the GST logger locations were largely unaffected by the disturbances. Conversely, a fence breach that was repaired in spring 2023 allowed scattered grazing in fenced plot A during the first part of our measurement period. While we do observe differences in vegetation cover between fenced plots A and B in both summer 2022 and 2023 (Figure 5a & c), these plots feature highly similar GSTs throughout the measurement period (Figure 13 and Table 6). Overall, the variability in GSTs among the fenced plots in Udleg is small compared to the difference between the fenced plots and the intensely grazed open plot.

## 5.3 Future research

The timing and intensity of grazing and trampling depends on the type and number of livestock, which is controlled by herding practices, such as seasonal movement of pastures. The seasonally contrasting effects of grazing on GSTs indicate that the seasonality of pasture use can be crucial for the net impact on GSTs. Only few studies have investigated how active herding and seasonal migrations localize grazing in time and space, but Lkhagvadorj et al. (2013) provide an overview from the Khangai Mountains (central Mongolia): flat grasslands in the mountain outskirts are the main grazing areas, while forest edges are grazed mostly at the end of the growing season. During autumn, winter and spring, herder families relocate livestock to drier areas where they have designated winter camps (Lkhagvadorj et al., 2013). Such preferential grazing of forest edges in autumn could allow grass to grow tall in summer while inhibiting buildup of litter layers prior to winter, leading to overall colder GSTs in these areas. Klinge et al. (2021) report permafrost extending several tens of meters into the steppe downslope of permafrost-underlain forest in the Khangai Mountains, which could be partially supported by such seasonal grazing patterns. However, it is unclear if this movement of pastures is also prevalent in our study region in the Khentii Mountains, as we observe livestock at our field sites also in winter. Furthermore, the composition of livestock in Mongolia has shifted towards more goats, which have the strongest impact on grassland ecosystems (Saizen et al., 2010). Such changes in livestock composition can possibly influence local grazing patterns, especially in winter when snow depth might limit smaller animals to areas that are more exposed to wind erosion or ablation such as ridges and south-facing slopes. Local herders in Udleg also report that smaller livestock such as sheep and goat are herded into enclosures overnight during winter, while horses, yak and cattle largely roam freely. If widespread, such active herding in winter will localize the impact these livestock have on both vegetation and snow cover. Understanding and quantifying the impact of spatiotemporal patterns of livestock activity on GSTs is beyond the scope of our study at this time and should be a topic for future research.

The observed differences in GST seasonality and dynamics have the potential to affect the subsurface thermal regime, including the presence of permafrost. In general, the thermal offset, i.e. the difference between MAGST and annual average temperature at the base of the seasonally frozen ground/top of permafrost, increases with increasing annual GST range (Smith & Riseborough, 1996). As we observe an amplified seasonal GST cycle in the intensely grazed open plots (Figure



12), deeper ground temperatures could be lowered below these plots compared to the adjacent fenced sites. However, livestock activity also limits the buildup of insulating soil organics (e.g. Gan et al., 2012; Odriozola et al., 2014) and leads to a drying of ground surface layers (e.g. Yan et al., 2018; Zhao et al., 2011), which is usually associated with a lower thermal offset (Romanovsky & Osterkamp, 1995; Smith & Riseborough, 1996). Due to these opposing effects, the net impact of grazing on the thermal offset is not clear.

Further studies could explore the potential of livestock management to lower ground temperatures at grassland sites, which could be a relevant strategy to protect marginal permafrost. Ishikawa et al. (2018) report degradation of permafrost across Mongolia in response to current warming, especially in low elevations and in grassland ecosystems, which coincides with the main rangelands. This study shows that excluding livestock leads to the formation of dense vegetation in grasslands, which lowers summertime GSTs by several degrees. This insight could be used to inform land use strategies, where e.g. animals are excluded from grasslands with marginal permafrost during the growing season while intense grazing or hay harvesting in autumn is used to limit the buildup of insulating litter layers and vegetation snow capture during the cold season. We find that combining GSTs from the fenced plot (April – October) with the GSTs from the open plot (November – March) would lower MAGSTs by 0.8-1.3°C in Udleg compared to the open and fenced plots, respectively. Such land use induced cooling of local GSTs could partially offset the 2.2°C increase of air temperatures observed across Mongolia (1940-2015; Zamba (2018)), potentially extending the climatic window for permafrost occurrence that is vital for downstream water supply and ecosystem function (Dashtseren et al., 2021; Ishikawa et al., 2018; Klinge et al., 2021). While the GST cooling effect of such grazing management or hay production would be greater on south-facing slopes (1.5-2.5°C in Terelj), these slopes are generally permafrost-free (e.g. Dashtseren et al. (2014), Ishikawa et al. (2005)).

Finally, the rate of carbon cycling in grasslands is greatly affected by grazing, which removes above-ground carbon and can drive long-term shifts in plant composition (Liang et al., 2021). The net effect of grazing on soil carbon stocks depends on factors such as precipitation, soil texture and dominant grass species, both an increase and a decrease in soil organic carbon with increasing grazing intensity has been reported (McSherry & Ritchie, 2013). In this study, we have studied GSTs at grazed and ungrazed plots, which can further affect the rates of soil carbon cycling. Higher ground temperatures generally stimulate soil microbial activity and associated soil respiration, increasing the release of carbon to the atmosphere (e.g. Schindlbacher et al., 2009). However, the effect of ground temperatures on soil carbon cycling can be limited by available soil moisture (e.g. Poll et al., 2013). We thus suggest future studies to investigate the rates of carbon uptake and release across the grazing gradients detailed in this study. Such investigations can reveal the potential impact of livestock management on the carbon balance of Mongolian grassland ecosystems.





**6 Conclusion**

In this study we investigate the variations of ground surface temperatures (GSTs) across local gradients of grazing intensity in the mountains of central Mongolia. We present 14 months of measured daily GSTs from fenced and open plots at two field sites with different topographic aspects and grazing history. These measurements are accompanied by surveys of snow and vegetation cover.

Grazed plots feature a substantially altered surface cover compared to ungrazed plots, with grazing leading to lower vegetation height and density, lower snow depths and no accumulation of surface litter layers. This leads to an intensified transfer of energy between the ground and atmosphere by reducing mechanisms such as canopy shading and insulation by snow and litter layers. Overall, we find a larger seasonal amplitude in GSTs at intensely grazed plots compared to ungrazed plots, with grazed plots being 1.4 to 5.1°C warmer in summer and -2.5 to -5.4°C colder in winter. The strongest difference in

GSTs between grazed and ungrazed plots are found on south-facing slopes, which we link directly, and indirectly through vegetation cover, to excess solar radiation at this site. However, the observed warming of grazed plots in summer is largely offset by colder wintertime GSTs, and the net effect of grazing on annual GSTs remains site specific.

Our findings suggest that local ground temperatures can be influenced by livestock management, which can be favourable

for e.g. protection of marginal permafrost. Colder GSTs could be achieved through shielding select areas from grazing in the growing season, while allowing or even promoting trampling and grazing in the snow season.

**Appendix A – Calculation of Leaf Area Index**

We derive the Leaf Area Index (*LAI*) from hemispherical photographs taken at the logger locations. The photographs are

taken using a Insta360 x2 (© Arashi Vision Inc.) camera that is placed horizontally below the vegetation at each location. The images thus cover the upwards looking 180° perspective, including all protruding vegetation elements. The processing of the images in order to derive the LAI is done in the software Hemisfer (Schleppi et al., 2007; Thimonier et al., 2010). To distinguish between pixels that are showing the sky or canopy we apply a threshold to the blue band of each image, which can most easily separate bright and blue sky from dark and green vegetation. Due to differences in lighting conditions

between images, the optimal threshold value had to be determined for each image individually. The *LAI* is subsequently calculated using the methodology by Lang (1987). As our sites are on gentle slopes (Table 2) and the hemispherical photographs typically feature large gaps in the canopy (Figure 4), we also apply the available corrections for slope (Schleppi et al., 2007) and canopy clumping (Chen & Cihlar, 1995). The resulting *LAI*s (and threshold values) for all images are presented in Zweigel (2024b).




**Appendix B – GST data for individual loggers**

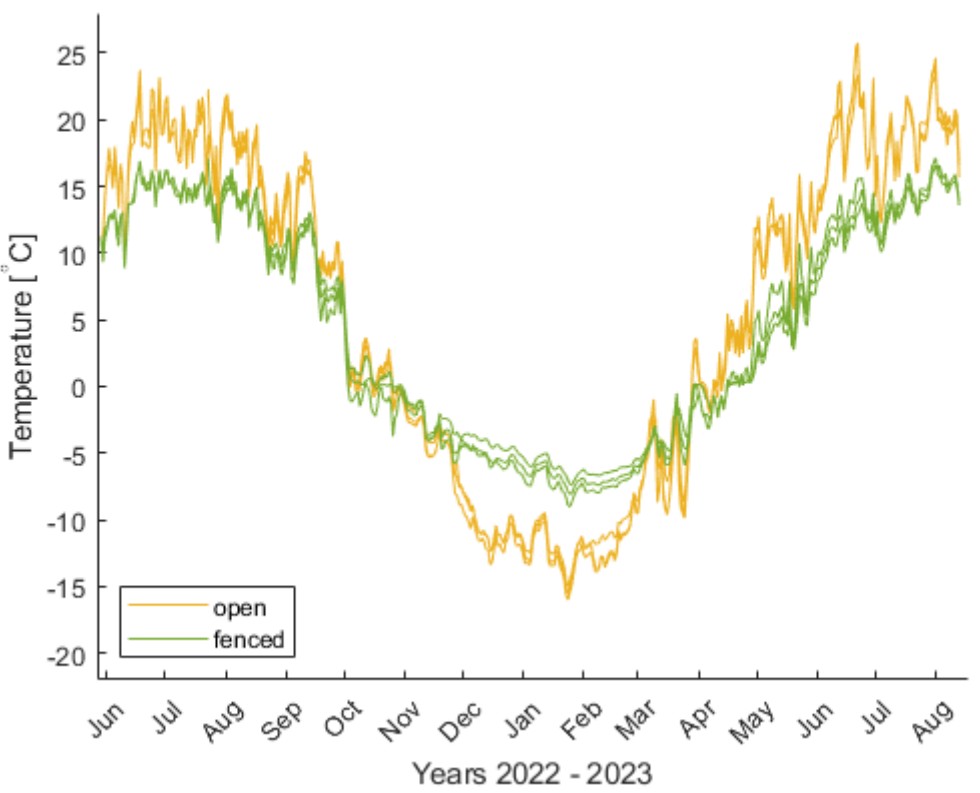

**Figure B1: As Figure 11, but showing the GST evolution of the individual loggers in Terelj. Note that two measurements per plot are available from 29. May 2022, while a third measurement was added on the 23. July 2022.**

**Table B1: As Table 5, but showing the GSTs of the individual loggers in Terelj. Loggers added on 23. July have insufficient data for summer (JJA) in 2022.**

| Logger | MAGST | JJA (2022) | SON | DJF | MAM | JJA (2023) |
|---|---|---|---|---|---|---|
| open 1 | 3.2 | 17.0 | 3.2 | -11.1 | 2.8 | 18.5 |
| open 2 | 2.8 | - | 2.6 | -12.1 | 2.7 | 18.8 |
| open 3 | 3.2 | 17.5 | 3.0 | -11.6 | 3.0 | 19.1 |
| fenced 1 | 2.3 | 13.7 | 2.7 | -6.3 | 0.0 | 12.7 |
| fenced 2 | 2.3 | 12.5 | 1.8 | -6.7 | 0.7 | 13.7 |
| fenced 3 | 2.6 | - | 2.1 | -5.5 | 0.5 | 13.4 |



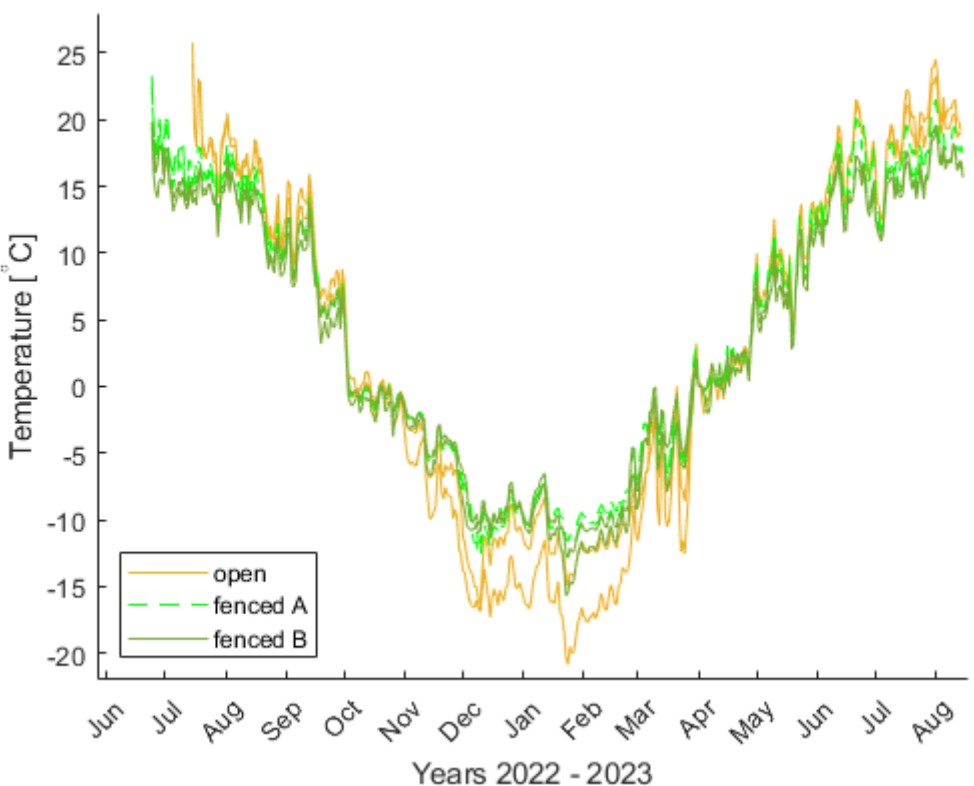

**Figure B2: As Figure 13, but showing the GST evolution of individual loggers in Udleg.**

**Table B2: As Table 6, but showing the GSTs of the individual loggers in Udleg. Note that the loggers in the open have insufficient measurements for JJA in 2022.**

| Logger | MAGST | JJA (2022) | SON | DJF | MAM | JJA (2023) |
|---|---|---|---|---|---|---|
| open east | 0.8 | - | 0.5 | -15.6 | 1.2 | 17.7 |
| open west | 2.5 | - | 1.9 | -11.7 | 2.0 | 18.5 |
| fenced A east | 2.2 | 15.1 | 1.5 | -9.6 | 1.9 | 15.5 |
| fenced A west | 2.8 | 15.7 | 1.7 | -9.5 | 2.6 | 17.0 |
| fenced B east | 1.8 | 13.7 | 0.9 | -9.7 | 1.2 | 15.3 |
| fenced B west | 2.0 | 14.3 | 1.4 | -10.5 | 2.3 | 15.0 |

## Data availability

The ground surface temperature data is available from Zweigel (2024a)

The snow and vegetation surveys are available from Zweigel (2024c)



The hemispherical photographs and calculated LAIs are available from Zweigel (2024b)

The ERA5 reanalysis data used to compute solar radiation is available from Hersbach et al. (2020).

**Author contribution**

RBZ and SW conceptualized the study. All authors contributed with data curation, while RBZ did the formal analysis, investigation and visualization. AD and SW administrated the project, and HL and SW acquired funding and contributed

with supervision. All authors contributed to the writing of the original manuscript.

**Competing interests**

The authors declare that they have no conflict of interest.

**Acknowledgements**

We thank Luc Girod for processing drone imagery from the Terelj site. This research has been supported by the Research

Council of Norway (Permafrost4Life, grant no. 301639; PRISM, grant no. 309625) and the European Space Agency CCI+ Permafrost (grant no. 4000123681/18/I-NB).

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
