# Peer review of "Impact of livestock activity on near-surface ground temperatures in Central Mongolian grasslands"

_EGUsphere, 2024_

## Author Comment (AC1)

**Author comment on Anonymous Referee #1**

*The above-mentioned article focuses on a topic that has received little attention to date, namely the influence of intensive grazing by mobile livestock farming (1), the effects on vegetation (2) and on the soil (3) as well as the resulting changes in permafrost (4). This is demonstrated in two local examples using series of measurements over a period of 14 months in two different exposures.*

We thank the referee for the positive feedback on our study and for taking the time to review our manuscript. We share the view that this topic has been currently understudied and feel that our study can provide relevant and novel insight and data. In this response, we have gone through the comments and suggestions made by the referee, which are shown in *blue italics*. Our response is given in normal font whereas our suggestions for the revised manuscript are provided in **bold text**.

*However, the title of the article is misleading in that the reader would expect generally valid statements for the whole of Mongolia. I would therefore strongly recommend adapting the title accordingly and also pointing out the limited duration of the measurements.*

We agree with the referee that the current title does not accurately reflect the spatial extent of our study. For the revised manuscript, we will adapt the title to "**Impact of livestock activity on near-surface ground temperatures in Central Mongolian grasslands**". Furthermore, we feel that the duration of our measurements is adequately stated in sentence 2 of the Abstract (line 13), as well as in our description of the study area (line 57). Indeed, a measurement period of 14 months can impact the validity of the findings especially if environmental conditions (such as precipitation) vary significantly from year to year. For this reason, we have included a discussion on this potential limitation (lines 440-444). In the revised manuscript, we will expand on this discussion by providing concrete examples of how our findings align with other studies covering multiple years by adding the following in line 441: **However, multi-year studies in grassland ecosystems show similar seasonal differences in GST as found in this study (Yan et al., 2018; Zhao et al., 2011). For example, Yan et al. (2018) observed summertime soil temperatures in heavily grazed plots on average 2,6°C warmer than at ungrazed plots across years with highly variable precipitation conditions. Furthermore, Zhang et al (2011) analysed 5 years of data from Inner Mongolia and found that denser vegetation cover at ungrazed sites provides insulation, resulting in colder summertime and warmer wintertime soil temperatures.**

*Does the paper address relevant scientific questions within the scope of BG? YES*

*Does the paper present novel concepts, ideas, tools, or data? NO*

We agree with the referee that several of the aspects our study build on previously established methods and ideas, and take the opportunity to clarify the novelty of our work. While the impact of grazing on ground surface temperatures has been previously studied, most studies use an experimental design where known animal loads are applied within enclosures (Wang et al., 2023; Yan et al., 2018, 2019; Zhao et al., 2011). Our study, on the other hand, investigates how traditional managed livestock affect ground surface temperatures, which includes how herders and animals seek to optimize the feeding conditions within the rangeland. Another novel aspect of our study is the role exposure has in determining surface cover and subsequently the local ground surface

temperature regime. In the revised manuscript we will clarify this study's original contribution by including the following:

Line 49: **In this study, we investigate how differences in livestock activity due to local fencing and semi-nomadic pastoralism affect vegetation, snow cover and associated GSTs, and how these manifests at sites with opposing topographic aspect. To achieve this we measured GSTs at multiple plots at two sites with different exposure and surveyed snow and vegetation cover in winter and summer.**

Line 314: **We find that in this landscape dominated by traditional pastoralism, the exclusion of grazing livestock allows for the establishment of higher and denser vegetation cover (Table 3 & 4), which leads to a dampened seasonal cycle in GST (Figure 11 & Figure 13). Furthermore, we find that topographic exposure shapes the local GST regime, with the largest differences in GST found at the south-facing Terelj site. Here, the grazed open plot is significantly warmer than the ungrazed fenced plot from April to September, while it is significantly colder in winter (Figure 12).**

*Are substantial conclusions reached? NO*

We thank the referee for this remark. In the revised manuscript, we will adjust the abstract and conclusion sections as to highlight the original conclusions enabled by this study.

Line 17: **We also find that the effect of grazing intensity depends on topographic aspect, with smaller seasonal differences of +1.4°C and -2.5°C found between grazed and ungrazed plots at a north-facing site. This relates to the lower available solar radiation at north-facing sites, which reduces the differences in vegetation cover between open and fenced plots.**

Line 534: **The difference in GSTs between grazed and ungrazed plots is strongly affected by topographic aspect, with the greatest difference found at the south-facing site. We link this directly, and indirectly through vegetation cover, to excess solar radiation at this site compared to the north-facing site.**

*Are the scientific methods and assumptions valid and clearly outlined? YES*

*Are the results sufficient to support the interpretations and conclusions? YES*

*Is the description of experiments and calculations sufficiently complete and precise to allow their reproduction by fellow scientists (traceability of results)? YES*

*Do the authors give proper credit to related work and clearly indicate their own new/original contribution? YES*

*Does the title clearly reflect the contents of the paper? NO*

We agree with the referee that the title needs revision and have suggested an adapted title in response to the referees comment above.

*Does the abstract provide a concise and complete summary? YES*

*Is the overall presentation well structured and clear? YES*

*Is the language fluent and precise? YES*

*Are mathematical formulae, symbols, abbreviations, and units correctly defined and used? YES*

*Should any parts of the paper (text, formulae, figures, tables) be clarified, reduced, combined, or eliminated? Partly, see my comment*

We have suggested amendments to the manuscript title to improve clarity.

*Are the number and quality of references appropriate? YES*

*Is the amount and quality of supplementary material appropriate? YES*

References

Wang, X., Zi, H., Wang, J., Guo, X., Zhang, Z., Yan, T., Wang, Q., & He, J.-S. (2023). Grazing-induced changes in soil microclimate and aboveground biomass modulate freeze–thaw processes in a Tibetan alpine meadow. *Agriculture, Ecosystems & Environment*, *357*, 108659. https://doi.org/10.1016/j.agee.2023.108659

Yan, Y., Yan, R., Chen, J., Xin, X., Eldridge, D. J., Shao, C., Wang, X., Lv, S., Jin, D., Chen, J., Guo, Z., Chen, B., & Xu, L. (2018). Grazing modulates soil temperature and moisture in a Eurasian steppe. *Agricultural and Forest Meteorology*, *262*, 157–165. https://doi.org/10.1016/j.agrformet.2018.07.011

Yan, Y., Yan, R., Wang, X., Xu, X., Xu, D., Jin, D., Chen, J., & Xin, X. (2019). Grazing affects snow accumulation and subsequent spring soil water by removing vegetation in a temperate grassland. *Science of The Total Environment*, *697*, 134189. https://doi.org/10.1016/j.scitotenv.2019.134189

Zhao, Y., Peth, S., Reszkowska, A., Gan, L., Krümmelbein, J., Peng, X., & Horn, R. (2011). Response of soil moisture and temperature to grazing intensity in a Leymus chinensis steppe, Inner Mongolia. *Plant and Soil*, *340*(1), 89–102. https://doi.org/10.1007/s11104-010-0460-9

---

## Author Comment (AC2)

**Author comment on Anonymous Referee #2**

*This paper discuss the grazing impacts on ground thermal conditions using onsite ground surface measurement, and snow and vegetation surveys. Writing style is well organized and text is understandable. Also the author's discussion and conclusion are consistent with the measurements, then I would recommend this acceptance after following revisions.*

We thank the referee for the positive evaluation of our manuscript. In this response, we have gone through the comments and suggestions made by the referee, which are shown in *blue italics*. Our response is given in normal font whereas our suggestions for the revised manuscript are provided in **bold text**.

*Methods*

*Resolution of GST measurements: the sensor accuracy is 0.5degree, is this enough to support your discussion?*

We agree with the referee that the numerical accuracy can be a limitation for the analysis and discussion of temperature difference below this accuracy. However, the main discussion topics and conclusions of our study relate to the differences in annual and seasonal (summer and winter) ground surface temperatures between our plots, which generally are larger than this numerical accuracy of the iButton temperature loggers (Table 5 and 6). Also, we limit the impact of individual logger measurements on our results by using daily averages of several logger measurements within each plot (See caption Tables 5 & 6 and Figures 11-13). In the revised manuscript we suggest to include the following clarification in line 123: **From the individual logger measurements, we calculate daily average GSTs within each plot, which we use throughout our analysis (Section 4.3).**

Furthermore, the 0.5°C numerical accuracy of the iButton temperature loggers can preclude analysis and discussion of phenomena that are confined to temperature intervals below this accuracy, i.e. the phase change of water. For this reason, we have refrained from discussing differences between the plots with respect to the timing of thawing and freezing throughout the manuscript.

*Your open site is just outside of the fence, where snow accumulation could be influenced wind disturbance of the fence and differ from site far from the fence.*

We thank the referee for this remark. Local disturbances of the wind field can indeed drive differential accumulation of snow, typically leading to preferentially deposition of snow on leesides. During our winter surveys at the Terelj and Udleg sites, we did observe snow wind drifts behind topographic features in the wider area. These drifts indicate the main wind direction to be along the main valley, which at both sited is oriented roughly East-West (Figure 1). For our study, we have however selected our plots to be adjacent to fences that run parallel to the main valley, and the effect of any preferential snow deposition should thus be limited. To clarify this in the revised manuscript, we will include the following text:

Line 80: **We note that the main wind direction in Terelj along of the main valley (East – West, Figure 1c), and that plots are placed either side of a fence that runs roughly parallel to the main valley (Figure 1a).**

Line 101: **Similar as at the Terelj site, the main wind direction in Udleg is along the main valley (East-Northeast – West-Southwest, Figure 1d), and both the open and fenced plots are placed so they avoid lee effects behind the adjacent fences (Figure 1b).**

*Can you show the location of vegetation, LAT and snow surveys on the map(Figure 1) ?*

We agree with the referee that specific information on the location of our snow, vegetation and LAI samples should be included in the manuscript. Indeed, our field data includes both samples collected in the immediate vicinity (<= 2m) of the temperature logger locations (Figure 1c & d), as well as surveys conducted in the near surroundings (<= 10m, within the respective fenced/open plots). No data was collected at the exact logger locations as to avoid any disturbances of the ongoing temperature measurements. Throughout our manuscript, we have used the term "survey" for data collected in the near surroundings of the logger locations:

- The survey of vegetation height and litter layer thickness/bare soil fraction at 30 points in the open and fenced plot in Terelj (Section 4.1.1, Figure 3)
- The survey of 100 snow depths and 25 snow densities within a visually undisturbed and a disturbed area in the open plot in Terelj (Section 4.2.1, Figure 7).

Conversely, the remaining field samples are taken in the immediate vicinity of the logger locations:

- All hemispherical images used to estimate LAI (Table 3, 4 & A1)
- The vegetation and litter heights reported for Udleg and Terelj, except for the abovementioned survey (Section 4.1.1 & 4.1.2, Table 3 & 4)
- All snow depths and densities reported fin Section 4.2.1 and 4.2.2, except for the abovementioned survey

To properly identify the location of our field samples in the revised manuscript we will:

-Mark the sections in which the snow and vegetation surveys were done in Figure 1c.

-Add the following text passage:

Line 151: **These measurements of vegetation and snow properties were done in the immediate vicinity of the GST logger location (Table 1, Figure 1), as to avoid disturbance of the ongoing measurements. In Terelj, we also conducted a more comprehensive survey of vegetation and snow cover in winter and summer of 2023 to capture the small-scale spatial variability in the near surroundings of the logger locations (Figure 1c).**

*Snow was measured only in one day. Did snowfall. clearing animal footprints, occurr before this day?*

We thank the referee for this remark. While we did only surveyed snow cover once, it is likely that areas trampled prior to previous snowfall events were covered with fresh snow. Indeed, our measurements of snow depths and bulk densities from fenced plot A in Udleg, where we observed extensive trampling by livestock (Figure 8a), suggest previous disturbances. Here, we observed a high variability of snow densities at locations with similar snow depths, ranging from ca. 110 to 300 kg/m$^3$ (Figure 9). These high densities could be the results of previous disturbances that compacted the snow cover, that since have been infilled during subsequent snowfall events. We discuss the effect of snow infilling of disturbed areas in lines 378-382 in the manuscript, while the effect of animal trampling on wintertime GSTs at our sites is discussed in line 429-437. Furthermore, we will include the following in the revised manuscript:

Line 248: **However, locations visually undisturbed at our time of visit could have experienced previous disturbances that since have been infilled with snow, which could lead to the large variability of snow densities in fenced plot B (Figure 9).**

*Results*

*Consider the difference in daily cycles of GST at fenced and open sites?*

We agree with the referee that the diurnal GST cycles are likely affected by the grazing intensity. However, the consideration of such short-term differences of GSTs is outside the scope of our study. Throughout the study we present daily average GSTs (Section 3.1), which we use to analyse and discuss grazing induced differences on monthly, seasonal, and annual timescales.

*L235 doubled 'Figure 7'* and *L248 doubled 'Figure9'*

We thank the referee for this remark and will correct these typos in the revised manuscript.

*Discussion*

*L313-320 This part is just summary of previous chapter, and could be shortened or omitted.*

We agree with the referee that this paragraph can be reduced, and suggest the following, shortened version for the revised manuscript:

**In this study, we use measured GSTs and observations of grassland vegetation and snow cover from the Khentii Mountains in Central Mongolia to quantify and compare the different temperature regimes across contrasts of grazing intensity and topography. Overall, we find that the exclusion of grazing livestock allows for higher and denser vegetation cover (Table 3 & 4) and a dampened seasonal cycle in GST (Figure 11 & Figure 13). The largest differences between intensely grazed and ungrazed plots are observed at the south-facing Terelj site, with monthly GST differences ranging from -5.8°C colder to 6.8°C (Figure 12). The grazing induced differences are are less pronounced at the north-facing Udleg site, where monthly GST differences range from -4.1°C to 3.6°C (Figure 12). The effect of intense grazing on MAGSTs is however small, with the open plot being 0.7°C warmer than the fenced plot in Terelj, while it is 0.4°C colder in Udleg (Table 5 and Table 6).**

*L335 could be 'Grazing and snow impcat on ground surface temperatures'*

We thank the referee for this remark. In section 5.1 "Grazing impact on ground surface temperatures" we do indeed also discuss how snow cover affects ground surface temperatures at our sites. The snow effects we discuss are however related to interactions with direct and indirect grazing effects such as: reduction of thermal insulation due to snow trampling by livestock (line 416-437), increased sublimation from disturbed snow surfaces (line 359-366), snow capture of standing litter at ungrazed sites (line 374-378), and snow infilling of disturbed areas by wind redistribution (line 378-383).

While studying the interactions between snow, terrain and vegetation at our highly continental sites would be interesting, it is outside the scope of the current study. We thus suggest keeping the original section heading also in the revised manuscript.

*L364-366 Do you have any image of snow condition such as onsite photo or satellite image?*

We agree with the referee that photographs of snow surface microtopography would strengthen the discussion. However, we were not successful in obtaining any photographs clearly showing the small-scale snow structures with the camera equipment available during our field visit. There is however available satellite imagery showing the intermittent nature of snow cover on steeper south-facing slopes. While we cite Sentinel-2 imagery (available through the Copernicus Browser; ESA (2024)), we suggest to include example images in the manuscript in the form of an appendix:

**Appendix C – Satellite imagery of snow conditions**

[Figure]

*Figure C1: Sentinel-2 L2A true color composite imagery of snow conditions in Udleg on a) 9.1.2023 and c) 27.1.2023, and in Terelj on b) 2.4.2023 and d) 7.4.2023. The imagery from Udleg shows how snow preferentially ablating from steep south-facing slopes even during the low solar radiation available in mid-winter. The Terelj imagery shows widespread ablation of snow on south-facing slopes in late winter. All imagery has the same scale and is downloaded through the Copernicus Browser (ESA, 2024).*

*L369-371 Unclear to see snow drift in the Fig 8.*

We thank the referee for this remark. Figure 8 does indeed not show wind drifts, but rather the classical features associated with wind erosion of the snow surface. We will clarify this in the revised manuscript by including the following modifications:

Line 251: **The variability in snow depths in the open is likely linked to local wind redistribution of snow, consistent with observed wind erosional features at the snow surface (Figure 8b).**

**Figure 8: Snow conditions in the Udleg study area on 25. February 2023. a) Fenced plot A (left of fence) has received less trampling than fenced plot B (right of fence). b) Open site with scattered livestock tracks and wind erosional features at the snow surface.**

Line 369: **In the open plot at Udleg, wind is channelled through the main valley, and we observe snow structures associated with wind erosion the measurement sites (Figure 8b), as well as deeper snow drifts in the surrounding.**

---

## Author Response (AR1)

Throughout this response, the referees' comments are shown in *blue italics*, our response is shown in normal font, and revised segments from the manuscript are shown in **bold font**.

All line numbers refer to the revised manuscript.

**Referee #1**

*The above-mentioned article focuses on a topic that has received little attention to date, namely the influence of intensive grazing by mobile livestock farming (1), the effects on vegetation (2) and on the soil (3) as well as the resulting changes in permafrost (4). This is demonstrated in two local examples using series of measurements over a period of 14 months in two different exposures.*

We thank the referee for the positive feedback on our study and for taking the time to review our manuscript. We share the view that this topic has been currently understudied and feel that our study can provide relevant and novel insight and data.

*However, the title of the article is misleading in that the reader would expect generally valid statements for the whole of Mongolia. I would therefore strongly recommend adapting the title accordingly and also pointing out the limited duration of the measurements.*

We agree with the referee that the current title does not accurately reflect the spatial extent of our study. For the revised manuscript, we have adapted the title to "**Impact of livestock activity on near-surface ground temperatures in Central Mongolian grasslands**". Furthermore, we feel that the duration of our measurements is adequately stated in sentence 2 of the Abstract (line 13), as well as in our description of the study area (line 59). Indeed, a measurement period of 14 months can impact the validity of the findings especially if environmental conditions (such as precipitation) vary significantly from year to year. For this reason, we included a discussion on this potential limitation. In the revised manuscript, we expand on this discussion by providing concrete examples of how our findings align with other studies covering multiple years by including the following:

Line 453: **However, multi-year studies in grassland ecosystems show similar seasonal differences in GST as found in this study (Yan et al., 2018; Zhao et al., 2011). For example, Yan et al. (2018) observed summertime soil temperatures in heavily grazed plots on average 2,6°C warmer than at ungrazed plots across years with highly variable precipitation conditions. Furthermore, Zhang et al (2011) analysed 5 years of data from Inner Mongolia and found that denser vegetation cover at ungrazed sites provides insulation, resulting in colder summertime and warmer wintertime soil temperatures.**

*Does the paper address relevant scientific questions within the scope of BG? YES*

*Does the paper present novel concepts, ideas, tools, or data? NO*

We agree with the referee that several of the aspects our study build on previously established methods and ideas, and take the opportunity to clarify the novelty of our work. While the impact of grazing on ground surface temperatures has been previously studied, most studies use an experimental design where known animal loads are applied within enclosures (Wang et al., 2023; Yan et al., 2018, 2019; Zhao et al., 2011). Our study, on the other hand, investigates how traditional managed livestock affect ground surface temperatures, which includes how herders and animals seek to optimize the feeding conditions within the rangeland. Another novel aspect of our study is the role exposure has in determining surface cover and subsequently the local ground surface temperature regime. In the revised manuscript we clarify this study's original contribution by including the following:

Line 50: **In this study, we investigate how differences in livestock activity due to local fencing and semi-nomadic pastoralism affect vegetation, snow cover and associated GSTs, and how these manifests at sites with opposing topographic aspect. To achieve this we measured GSTs at multiple plots at two sites with different exposure and surveyed snow and vegetation cover in winter and summer.**

Line 326: **Overall, we find that in this landscape dominated by traditional pastoralism, the exclusion of grazing livestock allows for higher and denser vegetation cover (Table 3 & 4) and a dampened seasonal cycle in GST (Figure 11 & Figure 13). Furthermore, we find that topographic exposure shapes the local GST regime with the largest differences between intensely grazed and ungrazed plots observed at the south-facing Terelj site.**

*Are substantial conclusions reached? NO*

We thank the referee for this remark. In the revised manuscript, we have adjusted the abstract and conclusion sections as to highlight the original conclusions enabled by this study.

Line 17: **We also find that the effect of grazing intensity depends on topographic aspect, with smaller seasonal differences of +1.4°C and -2.5°C found between grazed and ungrazed plots at a north-facing site. This relates to the lower available solar radiation at north-facing sites, which reduces the differences in vegetation cover between open and fenced plots.**

Line 550: **The difference in GSTs between grazed and ungrazed plots is strongly affected by topographic aspect, with the greatest difference found at the south-facing site. We link this directly, and indirectly through vegetation cover, to excess solar radiation at this site compared to the north-facing site.**

*Are the scientific methods and assumptions valid and clearly outlined? YES*

*Are the results sufficient to support the interpretations and conclusions? YES*

*Is the description of experiments and calculations sufficiently complete and precise to allow their reproduction by fellow scientists (traceability of results)? YES*

*Do the authors give proper credit to related work and clearly indicate their own new/original contribution? YES*

*Does the title clearly reflect the contents of the paper? NO*

We agree with the referee that the title needs revision and have suggested an adapted title in response to the referees comment above.

*Does the abstract provide a concise and complete summary? YES*

*Is the overall presentation well structured and clear? YES*

*Is the language fluent and precise? YES*

*Are mathematical formulae, symbols, abbreviations, and units correctly defined and used? YES*

*Should any parts of the paper (text, formulae, figures, tables) be clarified, reduced, combined, or eliminated? Partly, see my comment*

We have suggested amendments to the manuscript title to improve clarity.

*Are the number and quality of references appropriate? YES*

*Is the amount and quality of supplementary material appropriate? YES*

**Referee #2**

*This paper discuss the grazing impacts on ground thermal conditions using onsite ground surface measurement, and snow and vegetation surveys. Writing style is well organized and text is understandable. Also the author's discussion and conclusion are consistent with the measurements, then I would recommend this acceptance after following revisions.*

We thank the referee for the positive evaluation of our manuscript.

*Methods*

*Resolution of GST measurements: the sensor accuracy is 0.5degree, is this enough to support your discussion?*

We agree with the referee that the numerical accuracy can be a limitation for the analysis and discussion of temperature difference below this accuracy. However, the main discussion topics and conclusions of our study relate to the differences in annual and seasonal (summer and winter) ground surface temperatures between our plots, which generally are larger than this numerical accuracy of the iButton temperature loggers (Table 5 and 6). Also, we limit the impact of individual logger measurements on our results by using daily averages of several logger measurements within each plot (See caption Tables 5 & 6 and Figures 11-13). In the revised manuscript we have included the following clarification

Line 129: **From the individual logger measurements, we calculate daily average GSTs within each plot, which we use throughout our analysis (Section 4.3).**

Furthermore, the 0.5°C numerical accuracy of the iButton temperature loggers can preclude analysis and discussion of phenomena that are confined to temperature intervals below this accuracy, i.e. the phase change of water. For this reason, we have refrained from discussing differences between the plots with respect to the timing of thawing and freezing throughout the manuscript.

*Your open site is just outside of the fence, where snow accumulation could be influenced wind disturbance of the fence and differ from site far from the fence.*

We thank the referee for this remark. Local disturbances of the wind field can indeed drive differential accumulation of snow, typically leading to preferentially deposition of snow on leesides. During our winter surveys at the Terelj and Udleg sites, we did observe snow wind drifts behind topographic features in the wider area. These drifts indicate the main wind direction to be along the

main valley, which at both sited is oriented roughly East-West (Figure 1). For our study, we have however selected our plots to be adjacent to fences that run parallel to the main valley, and the effect of any preferential snow deposition should thus be limited. To clarify this in the revised manuscript, we include the following text:

Line 82: **We note that the main wind direction in Terelj is along of the main valley (East – West, Figure 1c), and that plots are placed either side of a fence that runs roughly parallel to the main valley (Figure 1a), avoiding potential lee effects behind the fence.**

Line 104: **Similar to Terelj site, the logger placement in Udleg avoids potential lee effects behind the fences, as the fences run roughly parallel to the main wind direction in Udleg which is along the main valley (East-Northeast – West-Southwest, Figure 1b, d).**

*Can you show the location of vegetation, LAT and snow surveys on the map(Figure 1) ?*

We agree with the referee that specific information on the location of our snow, vegetation and LAI samples should be included in the manuscript. Indeed, our field data includes both samples collected in the immediate vicinity (<= 2m) of the temperature logger locations (Figure 1c & d), as well as surveys conducted in the near surroundings (<= 10m, within the respective fenced/open plots). No data was collected at the exact logger locations as to avoid any disturbances of the ongoing temperature measurements. Throughout our manuscript, we have used the term "survey" for data collected in the near surroundings of the logger locations:

- The survey of vegetation height and litter layer thickness/bare soil fraction at 30 points in the open and fenced plot in Terelj (Section 4.1.1, Figure 3)
- The survey of 100 snow depths and 25 snow densities within a visually undisturbed and a disturbed area in the open plot in Terelj (Section 4.2.1, Figure 7).

Conversely, the remaining field samples are taken in the immediate vicinity of the logger locations:

- All hemispherical images used to estimate LAI (Table 3, 4 & A1)
- The vegetation and litter heights reported for Udleg and Terelj, except for the abovementioned survey (Section 4.1.1 & 4.1.2, Table 3 & 4)
- All snow depths and densities reported fin Section 4.2.1 and 4.2.2, except for the abovementioned survey

To properly identify the location of our field samples in the revised manuscript we have:

- Marked the sections in which the snow and vegetation surveys were done in Figure 1c)

[Figure]

**Figure 1: Overview showing the location of the temperature loggers in Terelj (a & c) and Udleg (b & d), and the location of our study sites within Mongolia (e). The location of our plots within each site is indicated by black text in c) and d), while the white arrow shows the location of the fence breach in 2022. a and b in c) indicate the snow surveys of areas with and without recent signs of trampling, respectively. All figures are oriented with North facing up, and a) and b) have the same scale. Background: a), b) and e) Satellite imagery (ESRI, 2023), c) and d) drone-based orthomosaic acquired on 17. and 15. August 2023, respectively, and processed in Agisoft Metashape (Agisoft LLC, 2023).**

- Added the following text passage:

Line 158: **These measurements of vegetation and snow properties were done in the immediate vicinity of the GST logger location (Table 1, Figure 1), as to avoid disturbance of the ongoing measurements. In Terelj, we also conducted a more comprehensive survey of vegetation and snow cover in winter and summer of 2023 to capture the small-scale spatial variability in the near surroundings of the logger locations. For these surveys, vegetation was surveyed at 30 points along lines in the open and fenced plots, while snow cover was surveyed at 100 points within two areas in the open plot (Figure 1c).**

*Snow was measured only in one day. Did snowfall. clearing animal footprints, occurr before this day?*

We thank the referee for this remark. While we did only surveyed snow cover once, it is likely that areas trampled prior to previous snowfall events were covered with fresh snow. Indeed, our

measurements of snow depths and bulk densities from fenced plot A in Udleg, where we observed extensive trampling by livestock (Figure 8a), suggest previous disturbances. Here, we observed a high variability of snow densities at locations with similar snow depths, ranging from ca. 110 to 300 kg/m³ (Figure 9). These high densities could be the results of previous disturbances that compacted the snow cover, that since have been infilled during subsequent snowfall events. We discuss the effect of snow infilling of disturbed areas in lines 390-394 in the manuscript, while the effect of animal trampling on wintertime GSTs at our sites is discussed in line 441-449. We have included the following in the revised manuscript:

Line 258: **However, locations visually undisturbed at our time of visit could have experienced previous disturbances that since have been infilled with snow, which could lead to the large variability of snow densities in fenced plot B (Figure 9).**

*Results*

*Consider the difference in daily cycles of GST at fenced and open sites?*

We agree with the referee that the diurnal GST cycles are likely affected by the grazing intensity. However, the consideration of such short-term differences of GSTs is outside the scope of our study. Throughout the study we present daily average GSTs (Section 3.1), which we use to analyse and discuss grazing induced differences on monthly, seasonal, and annual timescales.

*L235 doubled 'Figure 7'* and *L248 doubled 'Figure9'*

We thank the referee for this remark and have correct these typos in the revised manuscript.

*Discussion*

*L313-320 This part is just summary of previous chapter, and could be shortened or omitted.*

We agree with the referee that this paragraph can be reduced, and have included the following, shortened version for the revised manuscript:

Line 324: **In this study, we use measured GSTs and observations of grassland vegetation and snow cover from the Khentii Mountains in Central Mongolia to quantify and compare the different temperature regimes across contrasts of grazing intensity and topography. Overall, we find that in this landscape dominated by traditional pastoralism, the exclusion of grazing livestock allows for higher and denser vegetation cover (Table 3 & 4) and a dampened seasonal cycle in GST (Figure 11 & Figure 13). Furthermore, we find that topographic exposure shapes the local GST regime with the largest differences between intensely grazed and ungrazed plots observed at the south-facing Terelj site. Here, monthly GST differences range from -5.8°C colder to 6.8°C, whereas at the north-facing Udleg site, monthly GST differences range from -4.1°C to 3.6°C (Figure 12). The effect of intense grazing on MAGSTs is however small, with the open plot being 0.7°C warmer than the fenced plot in Terelj, while it is 0.4°C colder in Udleg (Table 5 and Table 6).**

*L335 could be 'Grazing and snow impcat on ground surface temperatures'*

We thank the referee for this remark. In section 5.1 "Grazing impact on ground surface temperatures" we do indeed also discuss how snow cover affects ground surface temperatures at our sites. The snow effects we discuss are however related to interactions with direct and indirect grazing effects such as: reduction of thermal insulation due to snow trampling by livestock (line 429-449), increased sublimation from disturbed snow surfaces (line 371-379), snow capture of standing

litter at ungrazed sites (line 386-389), and snow infilling of disturbed areas by wind redistribution (line 390-394).

While studying the interactions between snow, terrain and vegetation at our highly continental sites would be interesting, it is outside the scope of the current study. We thus keep the original section heading also in the revised manuscript.

*L364-366 Do you have any image of snow condition such as onsite photo or satellite image?*

We agree with the referee that photographs of snow surface microtopography would strengthen the discussion. However, we were not successful in obtaining any photographs clearly showing the small-scale snow structures with the camera equipment available during our field visit. There is however available satellite imagery showing the intermittent nature of snow cover on steeper south-facing slopes. While we cite Sentinel-2 imagery in the original manuscript (available through the Copernicus Browser; ESA (2024)), we have included example images in the revised manuscript in the form of an appendix:

**Appendix C – Satellite imagery of snow conditions**

[Figure]

*Figure C1: Sentinel-2 L2A true color composite imagery of snow conditions in Udleg on a) 9.1.2023 and c) 27.1.2023, and in Terelj on b) 2.4.2023 and d) 7.4.2023. The imagery from Udleg shows how snow preferentially ablating from steep south-facing slopes even during the low solar radiation available in mid-winter. The Terelj imagery shows widespread ablation of snow on south-facing slopes in late winter. All imagery has the same scale and is downloaded through the Copernicus Browser (ESA, 2024).*

*L369-371 Unclear to see snow drift in the Fig 8.*

We thank the referee for this remark. Figure 8 does indeed not show wind drifts, but rather the classical features associated with wind erosion of the snow surface. We will clarify this in the revised manuscript by including the following modifications:

Line 263: **The variability in snow depths in the open is likely linked to local wind redistribution of snow, consistent with observed wind erosion features at the snow surface (Figure 8b).**

**Figure 8: Snow conditions in the Udleg study area on 25. February 2023. a) Fenced plot A (left of fence) has received less trampling than fenced plot B (right of fence). b) Open site with scattered livestock tracks and wind erosion features at the snow surface.**

Line 381: **In the open plot at Udleg, wind is channelled through the main valley, and we observe snow structures associated with wind erosion the measurement sites (Figure 8b), as well as deeper snow drifts in the surrounding.**

**References**

ESA. (2024, March 11). *Copernicus Browser: Sentinel-2 L2A True color imagery*.

Wang, X., Zi, H., Wang, J., Guo, X., Zhang, Z., Yan, T., Wang, Q., & He, J.-S. (2023). Grazing-induced changes in soil microclimate and aboveground biomass modulate freeze–thaw processes in a Tibetan alpine meadow. *Agriculture, Ecosystems & Environment*, *357*, 108659. https://doi.org/10.1016/j.agee.2023.108659

Yan, Y., Yan, R., Chen, J., Xin, X., Eldridge, D. J., Shao, C., Wang, X., Lv, S., Jin, D., Chen, J., Guo, Z., Chen, B., & Xu, L. (2018). Grazing modulates soil temperature and moisture in a Eurasian steppe. *Agricultural and Forest Meteorology*, *262*, 157–165. https://doi.org/10.1016/j.agrformet.2018.07.011

Yan, Y., Yan, R., Wang, X., Xu, X., Xu, D., Jin, D., Chen, J., & Xin, X. (2019). Grazing affects snow accumulation and subsequent spring soil water by removing vegetation in a temperate grassland. *Science of The Total Environment*, *697*, 134189. https://doi.org/10.1016/j.scitotenv.2019.134189

Zhao, Y., Peth, S., Reszkowska, A., Gan, L., Krümmelbein, J., Peng, X., & Horn, R. (2011). Response of soil moisture and temperature to grazing intensity in a Leymus chinensis steppe, Inner Mongolia. *Plant and Soil*, *340*(1), 89–102. https://doi.org/10.1007/s11104-010-0460-9